

# Evaluating named entity recognition tools for extracting social networks from novels

Niels Dekker[1], Tobias Kuhn[1] and Marieke van Erp[2]

[1] Department of Computer Science, Vrije Universiteit Amsterdam, Amsterdam, The Netherlands
[2] DHLab, KNAW Humanities Cluster, Amsterdam, The Netherlands

## ABSTRACT

The analysis of literary works has experienced a surge in computer-assisted processing. To obtain insights into the community structures and social interactions portrayed in novels, the creation of social networks from novels has gained popularity. Many methods rely on identifying named entities and relations for the construction of these networks, but many of these tools are not specifically created for the literary domain. Furthermore, many of the studies on information extraction from literature typically focus on 19th and early 20th century source material. Because of this, it is unclear if these techniques are as suitable to modern-day literature as they are to those older novels. We present a study in which we evaluate natural language processing tools for the automatic extraction of social networks from novels as well as their network structure. We find that there are no significant differences between old and modern novels but that both are subject to a large amount of variance. Furthermore, we identify several issues that complicate named entity recognition in our set of novels and we present methods to remedy these. We see this work as a step in creating more culturally-aware AI systems.

## INTRODUCTION

The characters and their relations can be seen as the backbone of any story, and explicitly creating and analysing a network from these relationships can provide insights into the community structures and social interactions portrayed in novels (*Moretti, 2013*). Quantitative approaches to social network analysis to examine the overall structure of these social ties, are borrowed from modern sociology and have found their way into many other research fields such as computer science, history, and literary studies (*Scott, 2012*). *Elson, Dames & McKeown (2010)*, *Lee & Yeung (2012)*, *Agarwal, Kotalwar & Rambow (2013)*, and *Ardanuy & Sporleder (2014)* have all proposed methods for automatic social network extraction from literary sources. The most commonly used approach for extracting such networks, is to first identify characters in the novel through Named Entity Recognition (NER) and then identifying relationships between the characters through for example measuring how often two or more characters are mentioned in the same sentence or paragraph.

Many studies use off-the-shelf named entity recognisers, which are not necessarily optimised for the literary domain and do not take into account the surrounding cultural

Corresponding author
Marieke van Erp,
marieke.van.erp@dh.huc.knaw.nl

[1] We follow (*Sainte-Beuve, 1910*) here in defining a classic novel not as one written by the ancient Greeks or Romans ('the classics') but to canonical works.

context. Furthermore, to the best of our knowledge, such studies focus on social network extraction from 19th and early 20th century novels (which we refer to as *classic novels*).[1] Typically, these classic novels are obtained from Project Gutenberg (http://gutenberg.org/), where such public domain books are available for free. While beneficial for the accessibility and reproducibility of the studies in question, more recent novels may not imitate these classic novels with respect to structure or style. It is therefore possible that classic novels have social networks that have a structure that is very different from more recent literature. They might differ, for example, in their overall number of characters, in the typical number of social ties any given character has, in the presence or absence of densely connected clusters, or in how closely connected any two characters are on average. Moreover, changes along dimensions such as writing style, vocabulary, and sentence length could prove to be either beneficial or detrimental to the performance of natural language processing techniques. This may lead to different results even if the actual network structures remained the same. *Vala et al. (2015)* did compare 18th and 19th century novels on the number of characters that appear in the story, but found no significant difference between the two. Furthermore, an exploration of extracted networks can also be used to assess the quality of the extracted information and investigate the structure of the expression of social ties in a novel.

Thus far, we have not found any studies that explore how NER tools perform on a diverse corpus of fiction literature. In this study, we evaluate four different tools on a set of classic novels which have been used for network extraction and analyses in prior work, as well as more recent fiction literature (henceforth referred to as *modern novels*). We need such an evaluation to assess the robustness of these tools to variation in language over time (*Biber & Finegan, 1989*) and across literary genres. Comparing social networks extracted from corpora consisting of classic and modern novels may give us some insights into what characteristics of literary text may aid or hinder automatic social network extraction and provide indications of cultural change.

As previous work (*Ardanuy & Sporleder, 2014*) has included works from different genres, in this work we decided to focus on the fantasy/science fiction domain to smooth potential genre differences in our modern books. In our evaluation, we devote extra attention to the comparison between classic and modern fantasy/science fiction in our corpus.

We define the following research questions:

- *To what extent are off-the-shelf NER tools suitable for identifying fictional characters in novels?*
- *Which differences or similarities can be discovered between social networks extracted for different novels?*

To answer our first research question, we first evaluate four named entity recognisers on 20 classic and 20 modern fantasy/science fiction novels. In each of these novels, the first chapter is manually annotated with named entities and coreference relations. The named entity recognisers we evaluate are: (1) BookNLP (*Bamman, Underwood & Smith, 2014*;

2 A gazetteer is a list of names

https://github.com/dbamman/book-nlp—commit: 81d7a31) which is specifically tailored to identify and cluster literary characters, and has been used to extract entities from a corpus of 15,099 English novels. At the time of writing, this tool was cited 80 times. (2) Stanford NER version 3.8.0 (*Finkel, Grenager & Manning, 2005*), one of the most popular named entity recognisers in the NLP research community, cited 2,648 times at the time of writing. (3) Illinois Named Entity Tagger version 3.0.23 (*Ratinov & Roth, 2009*), a computationally efficient tagger that uses a combination of machine learning, gazetteers,[2] and additional features extracted from unlabelled data. At the time of writing, the system was downloaded over 10,000 times. Our last system (4) is IXA-Pipe-NERC version 1.1.1 (*Agerri & Rigau, 2016*), a competitive classifier that employs unlabelled data via clustering and gazetteers that outperformed other state-of-the-art NER tools on their within and out-domain evaluations.

To answer the second research question, we use the recognised named entities to create a co-occurrence network for each novel. Network analysis measures are then employed to compare the extracted networks from the classic and modern novels to investigate whether the networks from the different sets of novels exhibit major differences.

The contributions of this paper are: (1) a comparison and an analysis of four NER on 20 classic and 20 modern novels; (2) a comparison and an analysis of social network analysis measures on networks automatically extracted from 20 classic and 20 modern novels; (3) experiments and recommendations for boosting performance on recognising entities in novels; and (4) an annotated gold standard dataset with entities and coreferences of 20 classic and 20 modern novels.

The remainder of this paper is organised as follows. We first discuss related work in the section 'Related Work'. Next, we describe our approach and methods in the section 'Materials and Data Preparation'. We present our evaluation of four different NER systems on 20 classic and 20 modern novels in the section 'Named Entity Recognition Experiments and Results', followed by the creation and analysis of social networks in the section 'Network Analysis'. We discuss issues that we encountered in the identification of fictional characters and showcase some methods to boost performance in the section 'Discussion and Performance Boosting Options'. We conclude by suggesting directions for future work in the section 'Conclusion and Future Work'.

The code for all experiments as well as annotated data can be found at https://github.com/Niels-Dekker/Out-with-the-Old-and-in-with-the-Novel.

## RELATED WORK

As mentioned in the section 'Introduction', we have not found any other studies that compared the performances of social network extraction on classic and modern novels; or compared the structures of these networks. This section therefore focuses on the techniques used on classic literature. In first part of this section, we will describe how other studies extract and cluster characters. In the second part, we outline what different choices can be made for the creation of a network, and motivate our choices for this study.

## Named entity recognition

The first and foremost challenge in creating a social network of literary characters is identifying the characters. NER is often used to identify passages in text that identify things by a name. Furthermore, identified passages are often also classified into various categories such as *person*, *location*, and *organisation*. Typically, this approach is also used to identify miscellaneous numerical mentions such as dates, times, monetary values, and percentages.

*Elson, Dames & McKeown (2010)*, *Ardanuy & Sporleder (2014)*, *Bamman, Underwood & Smith (2014)*, and *Vala et al. (2015)* all use the Stanford NER tagger (*Finkel, Grenager & Manning, 2005*) to identify characters in literary fiction. On a collection of Sherlock Holmes novels, these studies perform Named entity recognition, $F_1$-scores between: 45 and 54. *Vala et al. (2015)* propose that the main difficulty with this collection is the multitude of minor characters, a problem which we expect to be also present in our collections of classic and modern novels.

A big difference between the news domain (for which most language technology tools have been created) and the literary domain, is that names do not have to follow the same 'rules' as names in the real world. This topic is explored in the Namescape project by *De Does et al. (2017)* namescape (http://blog.namescape.nl/). In this project, one million tokens taken from 550 Dutch novels were manually annotated. A distinction between first and last names was made in order to test whether different name parts are used with different effects. A named entity recogniser was trained specifically for this corpus by namescape-clin, obtaining an $F_1$ score of 93.60 for persons. The corpus contains fragments of novels written between the 17th and 20th century, but as the corpus and tools are not available, we cannot investigate its depth or compare it directly to our work. Other approaches attempt to use the identification of locations and physical proximity to improve the creation of a social network (*Lee & Yeung, 2012*).

## Coreference resolution

One difficulty of character detection is the variety of aliases one character might go by, or; coreference resolution. For example, George Martin's *Tyrion Lannister*, might alternatively be mentioned as *Ser Tyrion Lannister, Lord Tyrion, Tyrion, The Imp* or *The Halfman*. In the vast majority of cases, it is desirable to collapse those character references into one character entity. However, in some cases, retaining some distinction between character references can be useful: we provide an example of this in subsection 'Network Exploration'.

Two distinct approaches attempt to address this difficulty, (1) omit parts of a multi-word name, or (2) compile a list of aliases. The former approach leaves out honorifics such as the *Ser* and *Lord* in the above example in order to cluster the names of one character. To automate this clustering step, some work has been done by *Bamman, Underwood & Smith (2014)* and *Ardanuy & Sporleder (2014)*. While useful, the former approach alone provides no solace for the matching of the last two example aliases; where no part of the character's name is present. The latter approach thus suggests to manually compile a list of aliases for each character with the aid of external resources or annotators. This method is utilised by *Elson, Dames & McKeown (2010)* and

*Lee & Yeung (2012)*. In namescape-clin, wikification (i.e. attempting to match recognised names to Wikipedia resources) is used. Obviously this is most useful for characters that are famous enough to have a Wikipedia page. The authors state in their error analysis *Van Dalen-Oskam et al. (2014*, Section 3.2*)* that titles that are most likely from the fantasy domain are most difficult to resolve, which already hints at some differences between names in different genres.

### Anaphora resolution

To identify as many character references as possible, it is important to take into account that not all references to a character actually mention the character's name. In fact, *Bamman, Underwood & Smith (2014)* show that 74% of character references come in the form of a pronouns such as *he, him, his, she, her,* and *hers* in a collection of 15,099 English novels. To capture these references, the anaphoric pronoun is typically matched to its antecedent by using the linear word distance between the two, and by matching the gender of anaphora to that of the antecedent. The linear word distance can be, for example, the number of words between the pronoun and the nearest characters. For unusual names, as often found in science fiction and fantasy, identification of the gender may be problematic.

### Network creation

For a social network of literary characters, characters are represented by the nodes, whereas the edges indicate to some interaction or relationship. While the definition of a character is uniformly accepted in the literature, the definition of an interaction varies per approach. In previous research, two main approaches can be identified to define such an edge. On the one hand, **conversational networks** are used in approaches by *Chambers & Jurafsky (2008)*, *Elson & McKeown (2010),* and *He, Barbosa & Kondrak (2013)*. This approach focuses on the identification of speakers and listeners, and connecting each speaker and listener to the quoted piece of dialogue they utter or receive. On the other hand, **co-occurrence networks** (as used by *Ardanuy & Sporleder (2014)* and *Fernandez, Peterson & Ulmer (2015)*) are created by connecting characters if they occur in the same body of text. While conversational networks can provide a good view of who speaks directly to whom, *Ardanuy & Sporleder (2014)* argue that '*...much of the interaction in novels is done off-dialogue through the description of the narrator or indirect interactions*' (p. 34). What value to assign to the edges depends on the end-goal of the study. For example, *Fernandez, Peterson & Ulmer (2015)* assign a negative or positive sentiment score to the edges between each character-pair in order to ultimately predict the protagonist and antagonist of the text. *Ardanuy & Sporleder (2014)* used weighted edges to indicate how often two characters interact.

### Network analysis

Social network analysis draws upon network theory for its network analysis measures (*Scott, 2012*). The application of these measures to networks extracted from literature has been demonstrated insightful in assessing the relationships of characters in for example

'Alice in Wonderland' (*Agarwal et al., 2012*) and 'Beowulf', the 'Iliad' and 'Táin Bó Cuailnge' ('The Cattle Raid of Cooley', an Irish epic) (*Mac Carron & Kenna, 2012*). Network analysis can also play a role in authorship attribution (*Amancio, 2015*, *Akimushkin, Amancio & Oliveira, 2017*) and characterising a novel (*Elson, Dames & McKeown, 2010*).

## MATERIALS AND DATA PREPARATION

For the study presented here, we are interested in the recognition and identification of persons mentioned in classic and modern novels for the construction of the social network of these fictitious characters. We use off-the-shelf state-of-the-art entity recognition tools in an automatic pipeline without manually created alias lists or similar techniques. For the network construction, we follow *Ardanuy & Sporleder (2014)* and apply their co-occurrence approach for the generation of the social network links with weighted edges that indicate how often two characters are mentioned together. We leave the consideration of negative weights and sentiments for future work. Before we will explain the details of the used entity recognition tools, how they compare for the given task, and how their results can be used to build and analyse the respective social networks, we explain first the details of our selected corpus, how we preprocessed the data, and how we collected the annotations for the evaluation.

### Corpus selection

Our dataset consists of 40 novels—20 classic and 20 modern novels—the specifics of which are presented in Table A2 in the Appendix. Any selection of sources is bound to be unrepresentative in terms of some characteristics but we have attempted to balance breadth and depth in our dataset. Furthermore, we have based ourselves on selections made by other researchers for the classics and compilations by others for the modern books.

For the classic set, the selection was based on Guardian's Top 100 all-time classic novels (*McCrum, 2003*). Wherever possible, we selected books that were (1) analysed in related work (as mentioned in the subsection 'Coreference Resolution') and (2) available through Project Gutenberg (https://www.gutenberg.org/).

For the modern set, the books were selected by reference to a list compiled by BestFantasyBooksCom (http://bestfantasybooks.com/top25-fantasy-books.php, last retrieved: 30 October 2017). For our final selection of these novels, we deliberately made some adjustments to get a wider selection. That is, some of the books in this list are part of a series. If we were to include all the books of the upvoted series, our list would consist of only four different series. We therefore chose to include only the first book of each of such series. As the newer books are unavailable on Gutenberg, these were purchased online. These digital texts are generally provided in .epub or .mobi format. In order to reliably convert these files into plain text format, we used Calibre (https://calibre-ebook.com/—version 2.78), a free and open-source e-book conversion tool. This conversion was mostly without any hurdles, but some issues were encountered in terms of encoding, as is discussed in the next section. Due to copyright restrictions, we cannot share this full dataset but our gold standard

**Table 1 Annotation example.**

| Id | Preceding context | Focus sentence | Subsequent context | # | Person 1 | Person 2 |
|---|---|---|---|---|---|---|
| 541 | Bran reached out hesitantly | 'Go on', Robb told him | 'You can touch him' | 2 | Robb Stark | Bran Stark |

annotations of the first chapter of each are provided on this project's GitHub page. The ISBN numbers of the editions used in our study can be found in Table A2 the Appendix.

## Data preprocessing

To ensure that all the harvested text files were ready for processing, we firstly ensured that the encoding for all the documents was the same, in order to avoid issues down the line. In addition, all information that is not directly relevant to the story of the novel was stripped. Even while peripheral information in some books—such as appendices or glossaries—can provide useful information about character relationships, we decided to focus on the story content and thus discard this information. Where applicable, the following peripheral information was manually removed: (1) reviews by fellow writers, (2) dedications or acknowledgements, (3) publishing information, (4) table of contents, (5) chapter headings and page numbers, and (6) appendices and/or glossaries.

During this clean-up phase, we encountered some encoding issues that came with the conversion to plain text files. Especially in the modern novels, some novels used inconsistent or odd quotation marks. This issue was addressed by replacing the inconsistent quotation marks with neutral quotations that are identical in form, regardless of whether if it is used as opening or closing quotation mark.

## Annotation

Because of limitations in time and scope, we only annotated approximately one chapter of each novel. In this subsection, we describe the annotation process.

### Annotation data

To evaluate the performance for each novel, a gold standard was created manually. Two annotators (not the authors of this article) were asked to evaluate 10 books from each category. For each document, approximately one chapter was annotated with entity co-occurrences. Because the length of the first chapter fluctuated between 84 and 1,442 sentences, we selected an average of 300 sentences for each book that was close to a chapter-boundary. For example, for *Alice in Wonderland*, the third chapter ended on the 315th sentence, so the first three chapters were extracted for annotation. While not perfect, we attempted to strike a balance between comparable annotation lengths for each book, without cutting off mid-chapter.

### Annotation instructions

For each document, the annotators were asked to annotate each sentence for the occurrence of characters. That is, for each sentence, identify all the characters in it. To describe this process, an example containing a single sentence from *A Game of Thrones* is included in Table 1. The **id** of the sentence is later used to match the annotated sentence to

**Table 2 Annotation instructions.**

| Guideline | Example |
|---|---|
| Ignore generic pronouns | 'Everyone knows; **you** don't mess with **me**!' |
| Ignore exclamations | 'For Christ's sake!' |
| Ignore generic noun phrases | 'Bilbo didn't know what to tell **the wizard**' |
| Include non-human named characters | 'His name is **Buckbeak**, he's a hippogriph' |

**Note:**
Boldface indicates an entity mention.

its system-generated counterpart for performance evaluation. The **focus sentence** is the sentence that corresponds to this **id**, and is the sentence for which the annotator is supposed to identify all characters. As context, the annotators are provided with the **preceding** and **subsequent** sentences. In this example, the contextual sentences could be used to resolve the *'him'* in the **focus sentence** to *'Bran'*. To indicate how many persons are present, the annotators were asked to fill in the corresponding number (#) of people—with a maximum of 10 characters per sentence. Depending on this number of people identified, subsequent fields became available to the annotator to fill in the character names.

To speed up the annotation, an initial list of characters was created by applying the BookNLP pipeline to each novel. The annotators were instructed to map the characters in the text to the provided list to the best of their ability. If the annotator assessed that a person appears in a sentence, but is unsure of this character's identity, the annotators would mark this character as *default*. In addition, the annotators were encouraged to add characters, should they be certain that this character does not appear in the pre-compiled list, but occurs in the text nonetheless. Such characters were given a specific tag to ensure that we could retrieve them later for analysis. Lastly, if the annotator is under the impression that two characters in the list refer to the same person, the annotators were instructed to pick one and stick to that. Lastly, the annotators were provided with the peripheral annotation instructions found in Table 2.

While this identification process did include anaphora resolution of singular pronouns—such as resolving *'him'* to *'Bran'*—the annotators were instructed to ignore plural pronoun references. Plural pronoun resolution remains a difficult topic in the creation of social networks, as family members may sometimes be mentioned individually, and sometimes their family as a whole. Identifying group membership, and modelling that in the social network structure is not covered by any of the tools we include in our analysis or the related work referenced in the section 'Related Work' and therefore left to future work.

## NAMED ENTITY RECOGNITION EXPERIMENTS AND RESULTS

We evaluate the performance of four different NER systems on the annotated novels: BookNLP (*Bamman, Underwood & Smith, 2014*), Stanford NER (*Finkel, Grenager & Manning, 2005*), Illinois Tagger (*Ratinov & Roth, 2009*), and IXA-Pipe-NERC (*Agerri & Rigau, 2016*). The BookNLP pipeline uses the 2014-01-04 release of Stanford NER tagger (*Finkel, Grenager & Manning, 2005*) internally with the seven-class ontonotes model.

**Table 3 Precision (P), Recall (R), and $F_1$-scores of different NER systems on classic novels.**

| Title | BookNLP | | | Stanford NER | | | Illinois NER | | | IXA-NERC | | |
|---|---|---|---|---|---|---|---|---|---|---|---|---|
| | P | R | $F_1$ | P | R | $F_1$ | P | R | $F_1$ | P | R | $F_1$ |
| 1984 | **92.31** | 70.59 | 80.00 | **89.29** | 73.53 | 80.65 | **93.55** | 85.29 | **89.23** | **93.55** | 85.29 | **89.23** |
| A Study in Scarlet⊙ | 25.00 | 30.77 | 27.59 | 22.22 | 30.77 | 25.81 | 14.29 | 15.38 | 14.81 | 20.00 | 23.08 | 21.43 |
| Alice in Wonderland | 89.13 | 55.78 | 68.62 | 83.33 | 57.82 | 68.27 | 87.07 | 87.07 | 87.07 | 84.30 | 69.39 | 76.12 |
| Brave New World | 82.93 | 60.71 | 70.00 | 7.50 | 5.36 | 6.25 | 7.69 | 5.36 | 6.32 | 2.63 | 1.79 | 2.13 |
| David Copperfield⊙ | 29.41 | 35.71 | 32.26 | 54.02 | 67.14 | 59.87 | 58.82 | 71.43 | 64.52 | 14.47 | 15.71 | 15.07 |
| Dracula⊙ | 5.00 | 20.00 | 8.00 | 4.00 | 20.00 | 6.67 | 12.50 | 60.00 | 20.69 | 10.53 | 40.00 | 16.67 |
| Emma | 86.96 | 93.02 | 89.89 | 25.90 | 27.91 | 26.87 | 26.81 | 28.68 | 27.72 | 30.22 | 32.56 | 31.34 |
| Frankenstein⊙ | 52.00 | 76.47 | 61.90 | 37.93 | 64.71 | 47.83 | 30.77 | 47.06 | 37.21 | 34.62 | 52.94 | 41.86 |
| Huckleberry Finn | 86.84 | **98.51** | **92.31** | 81.08 | 89.55 | **85.11** | 77.92 | 89.55 | 83.33 | 79.71 | 82.09 | 80.88 |
| Dr. Jekyll and Mr. Hyde | 86.36 | 82.61 | 84.44 | 18.18 | 17.39 | 17.78 | 21.74 | 21.74 | 21.74 | 13.64 | 13.04 | 13.33 |
| Moby Dick⊙ | 67.65 | 74.19 | 70.77 | 63.89 | 74.19 | 68.66 | 68.42 | 83.87 | 75.36 | 37.84 | 45.16 | 41.18 |
| Oliver Twist | 85.61 | 94.44 | 89.81 | 36.30 | 42.06 | 38.97 | 44.32 | 33.62 | 38.24 | 34.69 | 40.48 | 37.36 |
| Pride and Prejudice | 79.26 | 94.69 | 86.29 | 32.33 | 38.05 | 34.96 | 29.37 | 32.74 | 30.96 | 33.87 | 37.17 | 35.44 |
| The Call of the Wild | 80.65 | 30.49 | 44.25 | 86.36 | 46.34 | 60.32 | 89.47 | 82.93 | 86.08 | 88.14 | 63.41 | 73.76 |
| The Count of Monte Cristo | 78.22 | 89.77 | 83.60 | 67.95 | 60.23 | 63.86 | 79.80 | 89.77 | 84.49 | 72.31 | 53.41 | 61.44 |
| The Fellowship of the Ring | 73.39 | 72.15 | 72.77 | 66.12 | 68.35 | 67.22 | 56.52 | 38.40 | 45.73 | 63.33 | 56.12 | 59.51 |
| The Three Musketeers | 65.71 | 29.49 | 40.71 | 63.64 | 35.90 | 45.90 | 45.45 | 25.64 | 32.12 | 73.68 | 35.90 | 48.28 |
| The Way We Live Now | 73.33 | 92.77 | 81.91 | 49.52 | 62.65 | 55.32 | 28.18 | 37.35 | 32.12 | 43.30 | 50.60 | 46.67 |
| Ulysses | 76.74 | 94.29 | 84.62 | 70.10 | **97.14** | 81.44 | 71.28 | **95.71** | 81.71 | 72.29 | **85.71** | 78.43 |
| Vanity Fair | 67.30 | 65.44 | 66.36 | 32.46 | 34.10 | 33.26 | 32.61 | 34.56 | 33.56 | 53.12 | 47.00 | 49.88 |
| Mean μ | 70.16 | 68.95 | 67.72 | 52.03 | 53.00 | 51.13 | 51.37 | 55.98 | 52.26 | 49.26 | 48.29 | 47.61 |
| Standard Deviation σ | 24.03 | 26.27 | 24.25 | 27.27 | 25.24 | 24.93 | 28.68 | 30.16 | 29.17 | 29.70 | 24.71 | 26.50 |

**Note:**
The highest scores in each column are highlighted in **bold**, and the lowest scores in *italics*. Novels written in 1st person are marked with ⊙.

As there have been several releases, and we focus on entities of type Person, we also evaluate the 2017-06-09 Stanford NER four-class CoNLL model.

The results of the different NER systems are presented in Table 3 for the classic novels, and Table 4 for the modern novels. All results are computed using the evaluation script used in the CoNLL 2002 and 2003 NER campaigns using the phrase-based evaluation setup (https://www.clips.uantwerpen.be/conll2002/ner/bin/conlleval.txt, last retrieved: 30 October 2017). The systems are evaluated according to micro-averaged precision, recall and $F_1$ measure. Precision is the percentage of named entities found by the system that were correct. Recall is the percentage of named entities present in the text that are retrieved by the system. The $F_1$ measure is the harmonic mean of the precision and recall scores. In a phrase-based evaluation setup, the system only scores a point if the complete entity is correctly identified, thus if in a named entity consisting of multiple tokens only two out of three tokens are correctly identified, the system does not obtain any points.

The BookNLP and IXA-Pipe-NERC systems require that part of speech tagging is performed prior to NER, we use the modules included in the respective systems

**Table 4 Precision (P), Recall (R), and $F_1$ scores of different NER systems on modern novels.**

| Title | BookNLP | | | Stanford NER | | | Illinois NER | | | IXA-NERC | | |
|---|---|---|---|---|---|---|---|---|---|---|---|---|
| | $P$ | $R$ | $F_1$ | $P$ | $R$ | $F_1$ | $P$ | $R$ | $F_1$ | $P$ | $R$ | $F_1$ |
| A Game of Thrones | **97.98** | 62.99 | 76.68 | 92.73 | 66.23 | 77.27 | **93.51** | 93.51 | **93.51** | 92.08 | 60.39 | 72.94 |
| Assassin's Apprentice⊙ | 63.33 | 38.38 | 47.80 | 61.19 | 41.41 | 49.90 | 61.45 | 40.40 | 48.78 | 53.12 | 34.34 | 41.72 |
| Elantris | 82.00 | 89.78 | 85.71 | 76.97 | 92.70 | 84.11 | 83.12 | **97.08** | 89.56 | 76.52 | 64.23 | 69.84 |
| Gardens of the Moon | *35.29* | *34.29* | *34.78* | 39.02 | 45.71 | 42.11 | *40.43* | 54.29 | 46.34 | 44.44 | 45.71 | 45.07 |
| Harry Potter | 83.80 | 90.36 | 86.96 | 61.24 | 65.66 | 63.37 | 58.43 | 58.43 | 58.43 | 54.94 | 53.61 | 54.27 |
| Magician | 72.92 | 42.17 | 53.44 | 65.57 | 48.19 | 55.56 | 77.67 | 96.39 | 86.02 | 63.10 | 63.86 | 63.47 |
| Mistborn | 96.46 | 81.95 | 88.62 | **93.22** | 82.71 | 87.65 | 90.07 | 95.49 | 92.70 | **94.05** | 59.40 | 72.81 |
| Prince of Thorns | 69.23 | 62.07 | 65.45 | 64.29 | 62.07 | 63.16 | 60.00 | 51.72 | 55.56 | 72.73 | 55.17 | 62.75 |
| Storm Front⊙ | 65.00 | 65.00 | 65.00 | 68.42 | 65.00 | 66.67 | 64.71 | 55.00 | 59.46 | 63.16 | 60.00 | 61.54 |
| The Black Company⊙ | 77.27 | 96.23 | 85.71 | *29.41* | *9.43* | *14.29* | 67.39 | 58.49 | 62.63 | 60.87 | *26.42* | *36.84* |
| The Black Prism | 90.29 | 90.29 | **90.29** | 88.35 | 88.35 | **88.35** | 88.68 | 91.26 | 89.95 | 87.21 | 72.82 | **79.37** |
| The Blade Itself | 62.50 | 71.43 | 66.67 | 71.43 | 71.43 | 71.43 | 52.63 | 71.43 | 60.61 | 55.56 | 35.71 | 43.48 |
| The Colour of Magic | 83.33 | 37.50 | 51.72 | 84.00 | 52.50 | 64.62 | 71.43 | *25.00* | *37.04* | 77.78 | 35.00 | 48.28 |
| The Gunslinger | 64.71 | **100.00** | 78.57 | 64.71 | **100.00** | 78.57 | 61.76 | 95.45 | 75.00 | 59.38 | **86.36** | 70.37 |
| The Lies of Locke Lamora | 86.16 | 74.05 | 79.65 | 87.58 | 76.22 | 81.50 | 86.79 | 74.59 | 80.23 | 88.19 | 68.65 | 77.20 |
| The Name of the Wind | 85.88 | 74.49 | 79.78 | 87.36 | 77.55 | 82.16 | 78.82 | 68.37 | 73.22 | 85.92 | 62.24 | 72.19 |
| The Painted Man | 87.02 | 71.70 | 78.62 | 86.47 | 72.33 | 78.77 | 80.81 | 87.42 | 83.99 | 83.09 | 71.07 | 76.61 |
| The Way of Kings | 80.72 | 87.01 | 83.75 | 75.82 | 89.61 | 82.14 | 70.10 | 88.31 | 78.16 | 66.67 | 49.35 | 56.72 |
| The Wheel of Time | 66.67 | 45.86 | 54.34 | 70.93 | 77.71 | 74.16 | 58.05 | 87.26 | 69.72 | 66.67 | 57.32 | 61.64 |
| Way of Shadows | 53.85 | 77.78 | 63.64 | 48.72 | 70.37 | 57.58 | 45.45 | 92.59 | 60.98 | *42.86* | 44.44 | 43.64 |
| Mean μ | 75.22 | 69.67 | 70.86 | 70.87 | 67.76 | 68.17 | 69.57 | 74.12 | 70.09 | 69.42 | 55.30 | 60.54 |
| Standard Deviation σ | 15.34 | 20.73 | 15.86 | 17.53 | 20.95 | 18.08 | 15.12 | 21.57 | 16.67 | 15.63 | 15.02 | 13.50 |

**Note:**
The highest scores in each column are highlighted in **bold**, and the lowest scores in *italics*. Novels written in 1st person are marked with ⊙.

for this. For Stanford NER and Illinois NE Tagger plain text is offered to the NER systems.

As the standard deviations on the bottom rows of Tables 3 and 4 indicate, the results on the different books vary greatly. However, the different NER systems generally do perform similarly on the same novels, indicating that difficulties in recognising named entities in particular books is a characteristic of the novels rather than the systems. An exception is *Brave New World* on which BookNLP performs quite well, but the others underperform. Upon inspection, we find that the annotated chapter of this book contains only five different characters among which 'The Director' which occurs 19 times. This entity is consistently missed by the systems resulting in a high penalty. Furthermore, the 'Mr.' in 'Mr. Foster' (occurring 31 times) is often not recognised as in some NE models titles are excluded. A token-based evaluation of Illinois NE Tagger on this novel for example yields a $F_1$-score of 51.91. The same issue is at hand with *Dr. Jekyll and Mr. Hyde* and *Dracula*. Although the main NER module in BookNLP is driven by Stanford NER, we suspect that additional domain adaptations in this package account for this performance difference.

When comparing the $F_1$-scores of the 1st person novels to the 3rd person novels in Tables 3 and 4, we find that the 1st person novels perform significantly worse than their 3rd person counterparts, at $p < 0.01$. These findings are in line with the findings of *Elson, Dames & McKeown (2010)*.

In the section 'Discussion and Performance Boosting Options', we delve further into particular difficulties that fiction presents NER with and showcase solutions that do not require retraining the entity models.

As the BookNLP pipeline in the majority of the cases outperforms the other systems and includes coreference resolution and character clustering, we further utilise this system to create our networks. The results of the BookNLP pipeline including the coreference and clustering are presented in Table A4. One of the main differences in that table is that if popular entities are not recognised by the system they are penalised heavier because the coreferent mentions are also not recognised and linked to the correct entities. This results in scores that are generally somewhat lower, but the task that is measured is also more complex.

## NETWORK ANALYSIS

In this section, we explain how the networks were created using the recognised named entities (subsection 'Network Construction'), followed by an explanation of network analysis measures that we applied to compare the networks (subsection 'Network Features'). We discuss the results of the analysis (subsection 'Results of Network Analysis'), as well as present an exploration of the network of one novel in particular to illustrate how a visualisation of a network can highlight particular characteristics of the interactions in the selected novel (subsection 'Network Exploration').

### Network construction

As explained in the section 'Related Work', we opt for the co-occurrence rather than the conversational method for finding the edges of our networks. The body of text that is used to define a co-occurrence differs per approach. Whereas *Fernandez, Peterson & Ulmer (2015)* define such a relation if characters are mentioned in the same sentence, *Ardanuy & Sporleder (2014)* use a paragraph for the same definition. We consider the delineation of what constitutes a paragraph to be too vague for the purpose of this study. While paragraphs are arguably better at conveying who interacts with whom, simply because of their increased length, it also brings forth an extra complexity in terms of their definition. Traditionally, paragraphs would be separated from another by means of a newline followed by an indented first line of the next paragraph. While this format holds for a part of our collection, it is not uniform. Other paragraph formats simply add vertical white space, or depend solely on the content (*Bringhurst, 2004*). Especially because the text files in our approach originate from different online sources— each with their own accepted format—we decided that the added ambiguity should be avoided. For this study, we therefore define that a co-occurrence relationship between two characters exists if they are mentioned in the same sentence. For a co-occurrence of more than two characters, we follow *Elson, Dames & McKeown (2010)*.

That is, a multi-way co-occurrence between four characters is broken down into six bilateral co-occurrences.

For the construction of each social network, the co-occurrences are translated to nodes for characters and edges for relationships between the characters. We thus create a static, undirected and weighted graph. For the weight of each edge, we follow *Ardanuy & Sporleder (2014)*. That is, each edge is assigned a weight depending on the number of interactions between two characters. For the construction of the network, we used NetworkX (https://networkx.github.io/—v1.11) and Gephi (https://gephi.org/—v0.9.1) to visualise the networks. To ground the network analysis to be presented below, we gathered some overall statistics of the network creation process shown in Table A3 on page 23. As mentioned in the subsection 'Annotation', if the annotator decided that a character was definitely present, but unable to assert which character, the occurrence was marked as *default*. The fraction of defaults represents what portion of all identified characters was marked with *default*. The fraction of unidentified characters represents the percentage of characters that were not retrieved by the system, but had to be added by the annotators. Next, we present some overall statistics such as sentence length, the average number of persons in a sentence, and the average fraction of sentences that mention a person. Lastly, we kept track of the total number of annotated sentences, the total number of unique characters and character mentions. The only difference that could be identified between classes is the average sentence length, which was significant at $p < 0.01$. The sentences in classic books are significantly longer than in modern novels, suggesting that there is indeed some difference in writing style. However, other than that, none of the other measures differ significantly. This is useful information, as it helps support that the novels used in either class are comparable, despite their age-gap.

## Network features

We analyse the following eight network features:

(1) **Average degree** is the mean degree of all the nodes in the network. The degree of a node is defined as the number of other nodes the node is connected to. If the degree of a node is zero, the node is connected to no other nodes. The degree of a node in a social network is thus is measure of its social 'activity' (*Wasserman & Faust, 1994*). A high value—for example, in *Ulysses*—indicates that the characters interact with many different other characters. Contrarily, a low value—for example, in *1984*—indicates that the characters only interact with a small number of other characters.

(2) **Average Weighted Degree** is fairly similar to the average degree, but especially in the sense of social networks, a distinction must be made. It differs in the sense that the weighted degree takes into account the weight of each of the connecting edges. Whereas a character in our social network could have a high degree—indicating a high level of social activity—if the weights of all those connected edges are relatively small, this suggests only superficial contact. Conversely, while the degree of a character could be low—for example, the character is only connected to two other characters— the two edges could have very large weights, indicating a deep social connection

between the characters. *Newman (2006)* underlines the importance of this distinction in his work on scientific collaborations. To continue the examples of *Ulysses* and *1984*; while their average degrees are vastly different (with Ulysses being the highest of its class and 1984 the lowest), their average *weighted* degrees are comparable.

(3) **Average Path Length** is the mean of all the possible shortest paths between each node in the network; also known as the geodesic distance. If there is no path connecting two nodes, this distance is infinite and the two nodes are part of different graph components (see item 7, Connected Components). The shortest path between two nodes can be found by using Dijkstra's algorithm (*Dijkstra, 1959*). The path length is typically an indication of how efficiently information is relayed through the network. A network with a low path length would indicate that the people in the network can reach each other through a relatively small number of steps.

(4) **Network Diameter** is the longest possible distance between two nodes in the network. It is in essence the longest, shortest path that can be found between any two nodes in the network, and is indicative of the linear size of the network (*Wasserman & Faust, 1994*).

(5) **Graph density** is the fraction of edges compared to the total number of possible edges. It thus indicates how complete the network is, where completeness would constitute all nodes being directly connected by an edge. This is often used in social network analysis to represent how closely the participants of the network are connected (*Scott, 2012*).

(6) **Modularity** is used to represent community structure. The modularity of a network is '...*the number of edges falling within groups minus the expected number in an equivalent network with edges placed at random*' (*Newman, 2006*). Newman shows modularity can be used as an optimisation metric to approximate the number of community structures found in the network. To identify the community structures, we used the Louvain algorithm (*Blondel et al., 2008*). The identification of community structures in graph is useful, because the nodes in the same community are more likely to have other properties in common (*Danon et al., 2005*). It would therefore be interesting to see if differences can be observed between the prevalence of communities between the classic and modern novels.

(7) **Connected components** are the number of distinct graph compartments. That is, a graph component is a subgraph in which any two vertices are connected to each other by paths, and which is connected to no additional vertices in the supergraph. In other words, it is not possible to traverse from one component to another. In most social communities, one 'giant component' can typically be identified, which contains the majority of all vertices (*Kumar, Novak & Tomkins, 2010*). A higher number of connected components would indicate a higher number of isolated communities. This is different from modularity in the sense that components are more strict. If only a single edge goes out from a subgraph to the supergraph, it is no longer considered a separate component. Modularity attempts to identify those communities that are basically 'almost' separate components.

(8) **Average clustering coefficient** is the mean of all clustering coefficients. The clustering coefficient of a node can perhaps best be described as 'all-my-neighbours-know-each-other'.

Social networks with a high clustering coefficient (and low average path length) may exhibit **small world** (https://en.wikipedia.org/wiki/Smallworld_experiment) properties (*Watts & Strogatz, 1998*). The small world phenomenon was originally described by Stanley Milgram in his perennial work on social networks (*Travers & Milgram, 1967*).

## Results of network analysis

To answer our second research question, we compared the network features presented in the subsection 'Network Features' for the social networks of the two different sets of novels. Table A5 on page 25 shows the results. The most striking feature of these results is the wide variance across social networks on all these network measures for both the classic and the modern novels. The size of these network ranges from just 10 nodes to networks more than 50 times as large. The network size alone can also explain at least a large part of the differences in graph density, diameter, and average path length, but also average degree and clustering coefficient show wide variation.

While we can observe large variation overall, there is no clear difference between the two classes, that is, between classic and modern novels. None of the evaluated network features differ significantly between these classes. Graph density is the feature that comes closest to being significant ($p = 0.09$), with our classic novels on average exhibiting denser networks than the modern ones.

In order to better interpret these values, and in order to find out whether this variance in network features is by itself a characteristic property of social networks exposed in novels, or whether this is true for social networks in general, we need a point for comparison. For that purpose, we compare our network results to metrics that have been reported for other social network in the literature. Table 5 shows 10 such networks for comparison, including three small networks on karate club members, football players, and email users (*Telesford et al., 2011*), three medium-sized networks of mathematicians, a larger group of email users, and actors (*Boccaletti et al., 2006*), and four large networks of online platforms (*Mislove et al., 2007*).

We can see that social networks reported elsewhere exhibit a wide variation as well, showing (unsurprisingly) an even much wider range for the network size, with the reported online social networks reaching millions of nodes. Our networks from novels are on the lower end of the size range, with the smallest ones being smaller than the smallest network of our comparison set (Karate). This directly explains why the path lengths are also on the lower end of the range, but with a considerable overlap. With respect to the average degree, our novel networks are covered by the range given by these comparison networks, with even the outliers of our dataset being less extreme than the most extreme cases of the comparison networks. The same holds for the clustering coefficient, except for the outlier for a very small network with a clustering coefficient of 0 (Alice in Wonderland). In summary, we can say that social networks from novels appear to be no different than social networks in general in showing a high variation in basically all network features across different networks. While networks differ much individually, there is no significant fundamental difference between classic and modern novels.

**Table 5 Comparison to other social networks.**

| Network | via | Nodes | Average degree | Clustering coefficient | Average path length |
|---|---|---|---|---|---|
| Karate | *Telesford et al. (2011)* | **35**[†] | 4.46 | 0.55 | **2.41**[†] |
| Football | *Telesford et al. (2011)* | 115 | 10.66 | 0.40 | 2.51 |
| E-mail | *Telesford et al. (2011)* | 1,133 | 9.62 | 0.22 | 3.60 |
| Math1999 | *Boccaletti et al. (2006)* | 58,516 | 5.00 | 0.15 | **8.46**[◇] |
| e-mail | *Boccaletti et al. (2006)* | 59,812 | 2.88 | **0.03**[†] | 4.95 |
| Actors | *Boccaletti et al. (2006)* | 225,226 | **61.00**[◇] | **0.79**[◇] | 3.65 |
| YouTube | *Mislove et al. (2007)* | 1,157,827 | 1.81 | 0.14 | 5.10 |
| Flickr | *Mislove et al. (2007)* | 1,846,198 | 1.76 | 0.31 | 5.67 |
| Orkut | *Mislove et al. (2007)* | 3,072,441 | **1.50**[†] | 0.17 | 4.25 |
| LiveJournal | *Mislove et al. (2007)* | **5,284,457**[◇] | 1.62 | 0.33 | 5.88 |
| | *maximum* | 522 | 15.77 | 0.81 | 3.33 |
| Classic novels | *mean* | 106 | 6.14 | 0.60 | 2.49 |
| | *minimum* | 10 | 1.66 | 0.00 | 1.53 |
| | *maximum* | 314 | 10.50 | 0.75 | 4.06 |
| Modern novels | *mean* | 99 | 5.50 | 0.56 | 2.68 |
| | *minimum* | 27 | 3.00 | 0.42 | 2.22 |

**Notes:**
The highest scores in each column are highlighted with a ◇ and the lowest scores with a † for the comparison networks.

## Network exploration

In addition to the formal analysis above, we show here a more informal exploration of one of the networks in order to give a more intuitive explanation of our results. For that purpose, we selected the largest network of the modern novels, which is *A Game of Thrones*. A visualisation of that network is shown in Fig. 1. We see that it is a quite dense network with many connections (it has the highest average degree of all modern novels; see Table A5) and a complex structure. Despite this complexity, the relationship between the main characters of this novel can easily be identified from this visualisation, and one can clearly identify social clusters. Such informal visual explorations should then of course be substantiated with formal analyses, that is, by ranking the edges of the network by their weights and by applying a clustering algorithm in the case of the two given examples. As the readers of this novel might have already spotted, *Dany* resides in a completely different part of the world in this novel, which explains her distance from rest of the network. Moreover, in *A Game of Thrones*, this character does not at any point physically interact with any of the characters in the larger cluster. This highlights a caveat of the use of co-occurrence networks over conversational networks. The character *Dany* does not truly interact with the characters of this main cluster, but is rather name-dropped in conversations between characters in that cluster. Her character 'co-occurs' with the characters that drop her name and edges are created to represent that.

To stick with the example of *Dany*, we can also identify two seemingly separate characters, *Dany* and *Daenerys Targaryen* in Fig. 1. These names actually refer to the same entity. As mentioned in the section 'Related Work', this issue may be addressed by creating

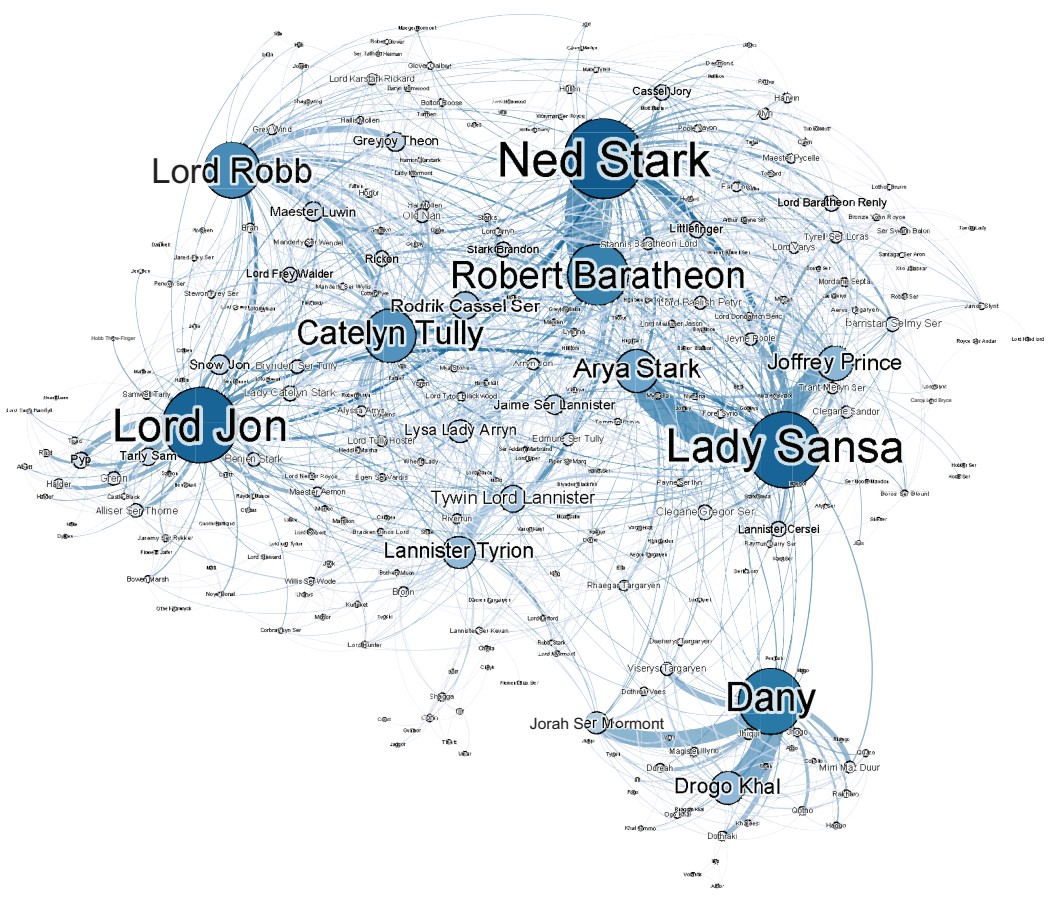

**Figure 1 Social network of G.R.R. Martin's *A Game of Thrones*.**

a list of aliases for each character. Some online sources exist that can help expedite this process, but we would argue these sources are not applicable to our modern novels. Whereas 19th century novels typically have characters with more traditional names such as *Elizabeth Bennet*, modern fantasy novels have unconventional names such as *Daenerys Targaryan*. External sources such as on metaCPAN[3] can help to connect *Elizabeth* to nicknames such as *Lizzy*, but there are no sources that can do this for *Daenerys* and *Dany*. Even if there was such a source, the question remains whether if it is desirable to collapse those characters. Especially in *A Game of Thrones*, the mentions of *Dany* and *Daenerys Targaryen* occur in entirely different contexts. Whereas references to *Dany* occur in an environment that is largely friendly towards her; her formal name of *Daenerys Targaryen* is mostly used by her enemies (in her absence). Rather than simply collapsing the two characters as one, it might be useful to be able to retain that distinction. This is a design choice that will depend on the type of research question one wants to answer by analysing the social networks.

## DISCUSSION AND PERFORMANCE BOOSTING OPTIONS

In analysing the output of the different NER systems, we found that some types of characters were particularly difficult to recognise. Firstly, we found a number of

[3] MetaCPAN is a search engine for Perl code and documentation: https://metacpan.org/source/BRIANL/Lingua-EN-Nickname-1.14/nicknames.txt (last retrieved: 30 October 2017).

**Table 6** Unidentified names in *The Black Company* replaced by generic English names.

| Original | Adjusted |
| --- | --- |
| Blue | Richard |
| Croaker | Thomas |
| Curly | Daniel |
| Dancing | Edward |
| Mercy | Charles |
| One-Eye | Timothy |
| Silent | James |
| Walleye | William |

unidentified names that are so called word names (i.e. terms that also occur in dictionaries, for example to denote nouns such as *Grace* or *Rebel*). We suspected that this might hinder the NER, which is why we collected all such names in our corpus in Table A1 on page 21, and highlighted such word names with a †. This table shows that approximately 50% of all unidentified names in our entire corpus consist at least partially of a word name, which underpins that this issue is potentially widely spread. In order to verify this, we replaced all potentially problematic names in the source material by generic English names. We made sure not to add names that were already assigned to other characters in the novel, and we ensured that these names were not also regular nouns. An example of these changed character names can be found in Table 6, which shows all names affected for *The Black Company*.

Secondly, we noticed that persons with special characters in their names can prove difficult to retrieve. For example, names such as *d'Artagnan* in *The Three Musketeers* or *Shai'Tan* in *The Wheel of Time* were hard to recognise for the systems. To test this, we replaced all names in our corpus such as *d'Artagnan* or *Shai'Tan* with *Dartagnan* and *Shaitan*. By applying these transformations to our corpus, we found that the performances could be improved, uncovering some of the issues that plague NER. As can be observed in Fig. 2, not all of the novels were affected by these transformations. Out of the 40 novels used in this study, we were able to improve the performance for 14. While the issue of the apostrophed affix was not as recurrent in our corpus as the real-word names, its impact on performance is troublesome nonetheless. Clearly, two novels are more affected by these transformations than the others, namely: *The Black Company* and the *The Three Musketeers*. To further sketch these issues, we delve a bit deeper into these two specific novels.

These name transformations show that the real-word names and names with special characters were indeed problematic and put forth a problem for future studies to tackle. As illustrated by Fig. 2, the aforementioned issues are also present in the classic novels typically used by related works (such as *The Three Musketeers*). This begs the question of the scope of these problems. To the best of our knowledge, similar works have not identified this issue to affect their performances, but we have shown that with a relatively simple workaround, the performance can be drastically improved. It would thus be interesting to evaluate how much these studies suffer from the same issue. Lastly, as

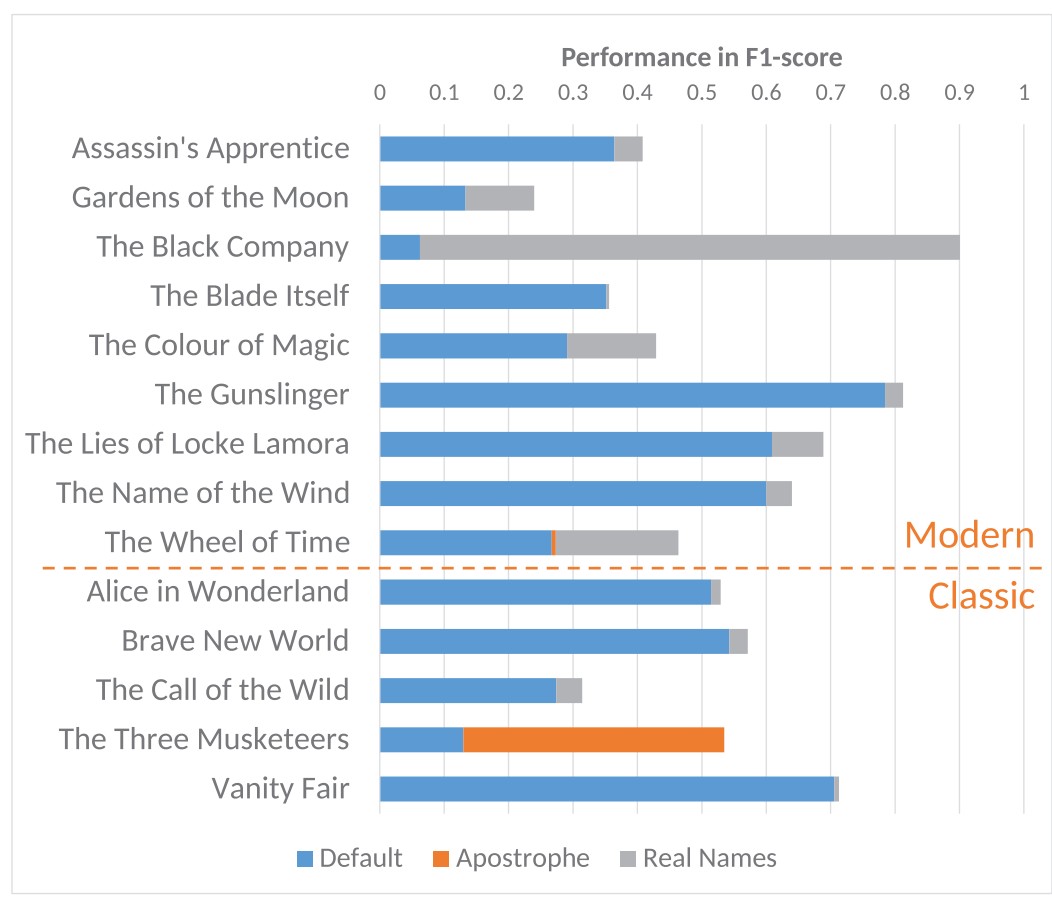

**Figure 2** Effect of transformations on all affected classic and modern novels in $F_1$-score in using the BookNLP pipeline (includes co-reference resolution).

manually replacing names is clearly far from ideal, we would like to encourage future work to find a more robust approach to resolve this issue.

## The Black Company

This fantasy novel describes the dealings of an elite mercenary unit—*The Black Company*—and its members, all of which go by code names such as the ones in Table 6. With a preliminary $F_1$-score of 06.85 (see Table A4), *The Black Company* did not do very well. We found this book had the highest percentage of unidentified characters of our collection. Out of the 14 characters found by our annotators, only five were identified by the pipeline. Interestingly enough, eight out of the nine unidentified characters in this novel have names that correspond to regular nouns. By applying our name transformation alone, the $F_1$-score rose from 06.85 to the highest in our collection to 90.00.

## The Three Musketeers

This classic piece recounts the adventures of a young man named *d'Artagnan*, after he leaves home to join the Musketeers of the Guard. With an $F_1$-score of 13.91 (see Table A4), *The Three Musketeers* performs the second worst of our corpus, and the worst in its class. By simply replacing names such as *d'Artagnan* with *Dartagnan* the $F_1$-score rose

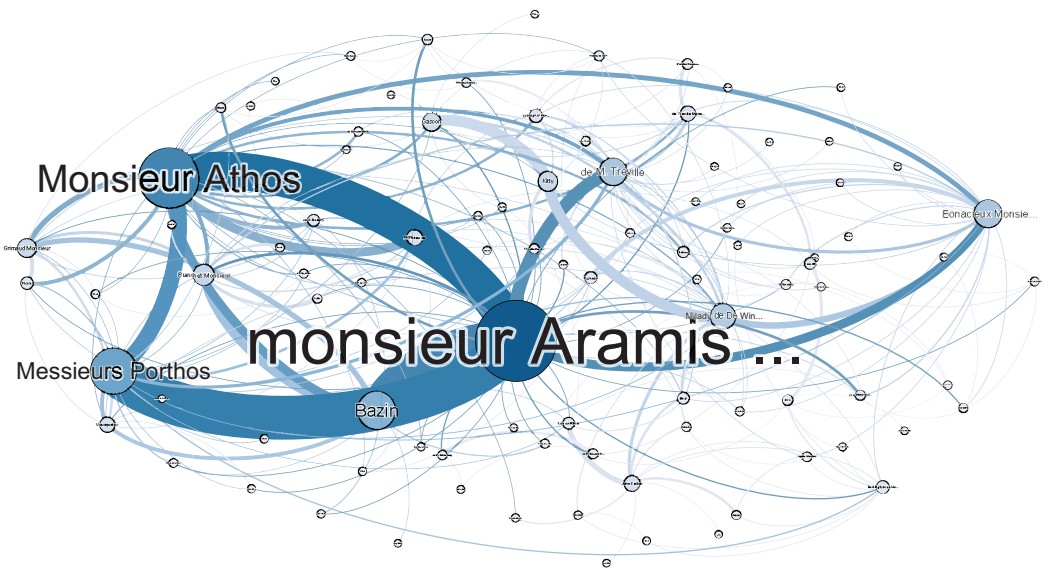

**Figure 3 Social network of *The Three Musketeers* without adjustment for apostrophed names.**

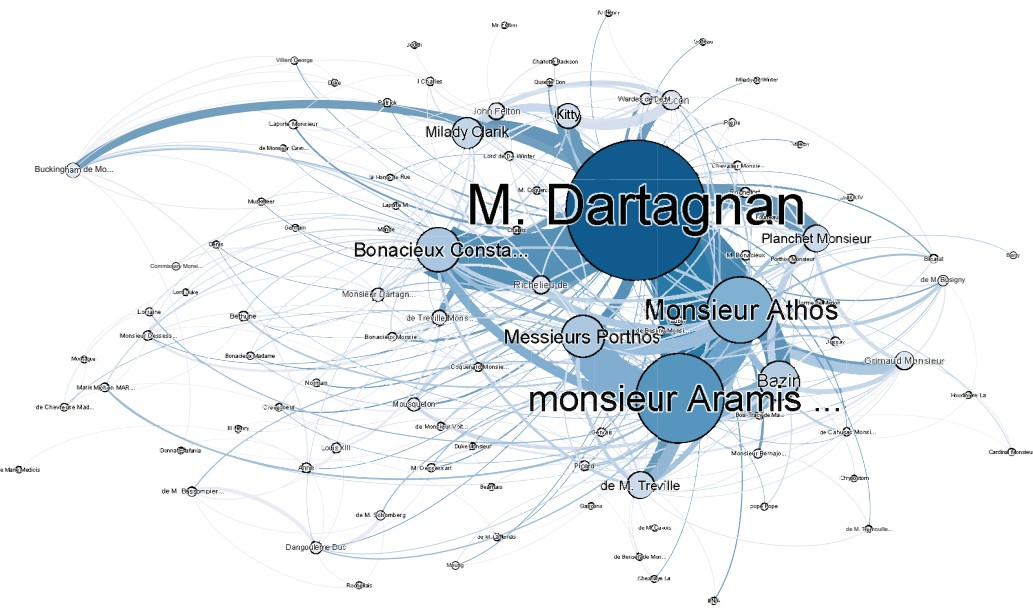

**Figure 4 Social network of *The Three Musketeers* with adjustment for apostrophed names.**

from 13.91 to 53, suggesting that the apostrophed name was indeed the main issue.
To visualise this, we have included figures of both *The Three Musketeer* networks—before and after the fix—in Figs. 3 and 4. As can be observed in Fig. 3, the main character of the novel is hardly represented in this network, which is not indicative of the actual story. The importance of resolving the issue of apostrophed named is made clear in Fig. 4, where the main character is properly represented.

## CONCLUSION AND FUTURE WORK

In this study, we set out to close a gap in the literature when it comes to the evaluation of NER for the creation of social networks from fiction literature. In our exploration of related work, we found no other studies that attempt to compare networks from classic and modern fiction. To fill this gap, we attempted to answer the following two research questions:

- *To what extent are off-the-shelf NER tools suitable for identifying fictional characters in novels?*
- *Which differences or similarities can be discovered between social networks extracted for different novels?*

To answer our primary research question, we evaluated four state-of-the-art NER systems on 20 classic and 20 modern science fiction/fantasy novels. In our study, we found no significant difference in performance of the named entity recognisers on classic novels and modern novels. We did find that novels written in 3rd person perspective perform significantly better than those written in 1st person, which is in line with findings in related studies. In addition, we observed a large amount of variance within each class, even despite our limitation for the modern novels to the fantasy/science fiction genre. We also identified some recurring problems that hindered NER. We delved deeper into two such problematic novels, and find two main issues that overarch both classes. Firstly, we found that word names such as *Mercy* are more difficult to identify to the systems. We showed that replacing problematic word names by generic placeholders can increase performance on affected novels. Secondly, we found that apostrophed names such as *d'Artagnan* also prove difficult to automatically identify. With fairly simple methods that capture some cultural background knowledge, we circumvented the above two issues to drastically increase the performance of the used pipeline. To the best of our knowledge, none of the related studies discussed in the section 'Related Work' acknowledge the presence of these issues. We would thus like to encourage future work to evaluate the impact of these two issues on existing studies, and call to develop a more robust approach to tackle them in future studies.

To answer our secondary research question, we created social networks for each of the novels in our collection and calculated several networks features with which we compared the two classes. As with the NER experiments, no major differences were found between the classic and modern novels. Again, we found that the distribution of network measures within a class was subject to high variance, which holds for our collection of both classic and modern novels. We therefore recommend that future work focuses on determining particular characteristics that can influence these analyses first and then perform a comparative analysis between subsets to see if this similarity between classes holds when the variance is reduced. Future studies could therefore attempt to compare classic and modern novels in the same genre or narration type (e.g. first-person vs third-person perspective). Lastly, different types of networks that for example collapse characters that occur under different names (cf. Dany and Daenerys) as well as dealing with plural pronouns and group membership (e.g. characters sometimes mentioned

individually and sometimes as part of a group) are currently unsolved problems for language technology and knowledge representation. These issues point to a strong need for more culturally-aware artificial intelligence.

## APPENDIX: ADDITIONAL STATISTICS

**Table A1** Characters that were not identified by the system, supplied by the annotators.

| Classic | | | Modern | |
|---|---|---|---|---|
| Ada | Howard | Mrs. Billington | Archmage of Ymitury† | Manie |
| Algy | Joanna | Mrs. Birch† | August† | Meena |
| Alice | Johnny | Mrs. Crisp† | Bil Baker† | Mercy† |
| Anna Boleyne | Jolly Miller† | Mrs. Effington Stubbs | Blue† | Mrs. Potter† |
| Aprahamian | Leonard | Mrs. Thingummy | Brine Cutter† | Old Cob† |
| Belisarius | Lord Mayor† | Murray | Bug† | One-Eye† |
| Best-Ingram | Lory† | Nathan Swain† | Chyurda | Pappa Doc† |
| Cain | Major Dover† | Peter Teazle† | Cotillion† | Patience† |
| Caroline | Marie Antoinette | Policar Morrel† | Croaker† | Plowman† |
| Catherine | Marshal Bertrand† | President West† | Curly† | Poul |
| Cato | Matilda Carbury | Queequeg | Dadda | Rand† |
| Cervantes | Matron† | Rip Van Winkle† | Dancing† | Shalash |
| Christine | Miss Birch† | Royce | Domi | Shrewd† |
| Chuck Loyola† | Miss Crump† | Sawbones† | Dow† | Silent† |
| Cleopatra | Miss Hopkins† | Semiramis | Elam Dowtry | Sirius† |
| Connolly Norman† | Miss King† | Shep | Elao | Talenel |
| Curly† | Miss Saltire† | Sir Carbury | Fredor | Talenelat |
| Dante | Miss Swindle† | Skrimshander† | Gart | Ted |
| Dave | Mme. D'Artagnan | Stamford | Harold | The Empress† |
| Dives† | Mollie | Stigand | Harvey | Themos Tresting |
| Dodo† | Mouse† | Sudeley | Howard | Theron |
| Dr. Floss† | Mr. Stroll† | Swubble | Ien | Threetrees |
| Duck† | Mr. Thursgood | The Director† | Ilgrand Lender† | Toffston |
| Edgar Atheling† | Mr. Beaufort† | Tommy Barnes | Ishar | Verus |
| Elmo | Mr. Crisp† | Unwin | Ishi | Walleye† |
| Farmer Mitchell† | Mr. Flowerdew | Ursula | Jim McGuffin† | Weasel† |
| Father Joseph† | Mr. Lawrence | Victor† | Kerible the Enchanter† | Willum |
| Fury† | Mr. Morris | Vilkins | Lilly† | Wit Congar† |
| Ginny | Mrs Loveday | Von Bischoff | | |
| Henry VIII | Mrs. Bates† | Ysabel | | |
| 39 out of 90 characters: 43% | | | 30 out of 56 characters: 54% | |

Note:
Characters whose names (partly) consist of a real word—such as 'Curly' or 'Mercy'—are marked with a †. Checked against http://dictionary.com.

**Table A2 Classic and modern novels included in this study.**

**Classic**

| Title | Author | (Year) | E-book No./ISBN |
|---|---|---|---|
| 1984 | George Orwell | (1949) | 9780451518651 |
| A Study in Scarlet | Conan Doyle | (1886) | 244 |
| Alice in Wonderland | Lewis Carroll | (1884) | 19033 |
| Brave New World | Aldous Huxley | (1865) | 9780965185196 |
| David Copperfield | Charles Dickins | (1931) | 766 |
| Dracula | Bram Stoker | (1850) | 345 |
| Emma | Jane Austen | (1897) | 158 |
| Frankenstein | Mary Shelley | (1815) | 84 |
| Huckleberry Finn | Mark Twain | (1818) | 76 |
| Jekyll and Hyde | Robert Stevenson | (1851) | 42 |
| Moby Dick | Herman Melville | (1838) | 2701 |
| Oliver Twist | Charles Dickins | (1813) | 730 |
| Pride and Prejudice | Jane Austen | (1886) | 1342 |
| The Call of the Wild | Jack London | (1903) | 215 |
| The Count of Monte Cristo | Alexandre Dumas | (1844) | 1184 |
| The Fellowship of the Ring | J. R. R. Tolkien | (1954) | 9780547952017 |
| The Three Musketeers | Alexandre Dumas | (1844) | 1257 |
| The Way We Live Now | Anthony Trollope | (1875) | 5231 |
| Ulysses | James Joyce | (1922) | 4300 |
| Vanity Fair | William Thackeray | (1847) | 599 |

**Modern**

| Title | Author | (Year) | E-book No./ISBN |
|---|---|---|---|
| A Game of Thrones | G.R.R. Martin | (1996) | 9780307292094 |
| Assassin's Apprentice | Robin Hobb | (1995) | 9781400114344 |
| Elantris | Brandon Sanderson | (2005) | 9780765383105 |
| Gardens of the Moon | Steven Erikson | (1999) | 9788498003178 |
| Harry Potter | J.K. Rowling | (1998) | 9781781103685 |
| Magician | Raymond Feist | (1982) | 9780007466863 |
| Mistborn | Brandon Sanderson | (2006) | 9788374805537 |
| Prince of Thorns | Mark Lawrence | (2011) | 9786067192681 |
| Storm Front | Jim Butcher | (2000) | 9781101128657 |
| The Black Company | Glen Cook | (1984) | 9782841720743 |
| The Black Prism | Brent Weeks | (2010) | 9782352945260 |
| The Blade Itself | Joe Abercrombie | (2006) | 9781478935797 |
| The Colour of Magic | Terry Pratchett | (1983) | 9788374690973 |
| The Gunslinger | Steven King | (1982) | 9781501143519 |
| The Lies of Locke Lamora | Scott Lynch | (2006) | 9780575079755 |
| The Name of the Wind | Patrick Rothfuss | (2007) | 9782352949152 |
| The Painted Man | Peter Brett | (2008) | 9780007518616 |
| The Way of Kings | Brandon Sanderson | (2010) | 9780765326355 |
| The Wheel of Time | Robert Jordan | (1990) | 9781857230765 |
| Way of Shadows | Brent Weeks | (2008) | 9781607513513 |

**Note:**
The short E-book numbers are the catalog entry of novels obtained from Gutenberg. Novels obtained through online purchase are denoted by the longer ISBNs.

**Table A3 Overall statistics for classic and modern novels in our corpus.**

**Classic**

| Title | Fraction of defaults | Fraction of unidentified characters | Average sentence length | Average persons per sentence | Fraction of sentences with a person | Annotated sentences | Unique characters | Total character mentions |
|---|---|---|---|---|---|---|---|---|
| 1984 | 0.55 | **0.00**[†] | 18.01 | 1.17 | 0.32 | 316 | 29 | 2,162 |
| A Study in Scarlet | 0.83 | 0.50 | 18.99 | 1.17 | 0.18 | 193 | 34 | 837 |
| Alice in Wonderland | 0.26 | **0.56**[°] | 20.99 | 1.23 | 0.79 | 316 | 17 | 656 |
| Brave New World | 0.35 | 0.17 | 15.87 | 1.06 | 0.25 | 299 | 51 | 1,809 |
| David Copperfield | 0.61 | **0.00**[†] | 22.79 | 1.08 | 0.49 | 261 | 157 | 9,922 |
| Dracula | **0.93**[°] | **0.00**[†] | 21.96 | **1.00**[†] | **0.06**[†] | 233 | 72 | 3,369 |
| Emma | 0.43 | 0.10 | 22.38 | 1.38 | 0.81 | 224 | 78 | 6,946 |
| Frankenstein | 0.86 | 0.22 | 25.80 | 1.19 | 0.17 | 300 | 29 | 658 |
| Huckleberry Finn | 0.59 | 0.14 | 23.46 | 1.20 | 0.40 | 215 | 82 | 1,749 |
| Jekyll and Hyde | 0.67 | 0.29 | 26.19 | 1.17 | 0.34 | **120**[†] | **13**[†] | **523**[†] |
| Moby Dick | 0.88 | 0.38 | 25.24 | 1.10 | 0.10 | 442 | 135 | 2,454 |
| Oliver Twist | 0.36 | 0.33 | 21.64 | 1.23 | 0.68 | 303 | 69 | 4,495 |
| Pride and Prejudice | 0.46 | 0.10 | 24.13 | 1.48 | 0.79 | 257 | 62 | 5,104 |
| The Call of the Wild | 0.49 | 0.50 | 21.67 | 1.31 | 0.61 | 192 | 28 | 731 |
| The Count of Monte Cristo | 0.47 | 0.25 | 21.91 | 1.35 | 0.79 | 197 | 250 | 13,562 |
| The Lord of the Rings | 0.47 | 0.48 | 16.30 | 1.20 | 0.46 | **769**[°] | 134 | 5,268 |
| The Three Musketeers | 0.60 | 0.36 | 19.19 | 1.13 | 0.49 | 265 | 115 | 4,842 |
| The Way We Live Now | 0.57 | 0.46 | 18.93 | 1.14 | 0.47 | 341 | 147 | **13,993**[°] |
| Ulysses | 0.57 | 0.33 | **13.35**[†] | 1.15 | 0.41 | 303 | **651**[°] | 8,510 |
| Vanity Fair | **0.24**[†] | 0.44 | **27.27**[°] | **1.54**[°] | **1.05**[°] | 256 | 359 | 11,503 |
| Mean μ | 0.56 | 0.28 | 21.30 | 1.21 | 0.48 | 290.10 | 125.60 | 4,954.65 |
| Standard Deviation σ | 0.20 | 0.18 | 3.67 | 0.14 | 0.27 | 131.89 | 150.20 | 4,403.32 |

**Modern**

| Title | Fraction of defaults | Fraction of unidentified characters | Average sentence length | Average persons per sentence | Fraction of sentences with a person | Annotated sentences | Unique characters | Total character mentions |
|---|---|---|---|---|---|---|---|---|
| A Game of Thrones | 0.29 | **0.00**[†] | 14.53 | 1.30 | **0.82**[°] | 283 | **322**[°] | **15,839**[°] |
| Assassin's Apprentice | 0.71 | 0.29 | 14.94 | 1.18 | 0.38 | 460 | 66 | 2,857 |
| Elantris | 0.32 | 0.27 | 14.24 | 1.10 | 0.60 | 367 | **14**[†] | **226**[†] |
| Gardens of the Moon | 0.75 | 0.44 | 12.20 | **1.03**[†] | 0.25 | 304 | 111 | 4,479 |
| Harry Potter | 0.32 | 0.33 | 15.55 | 1.33 | 0.74 | 338 | 84 | 5,114 |
| Magician | 0.49 | 0.17 | 14.78 | 1.16 | 0.45 | 310 | 115 | 4,976 |
| Mistborn | 0.34 | 0.22 | 12.90 | 1.19 | 0.68 | 297 | 104 | 11,672 |
| Prince of Thorns | 0.54 | **0.00**[†] | 12.33 | 1.14 | 0.38 | 107 | 79 | 2,282 |
| Storm Front | 0.77 | **0.00**[†] | 14.02 | 1.05 | 0.18 | 211 | 43 | 2,368 |
| The Black Company | 0.56 | **0.64**[°] | **9.73**[†] | 1.07 | 0.26 | 305 | 42 | 1,908 |
| The Black Prism | 0.50 | 0.14 | 13.19 | 1.04 | 0.40 | 380 | 88 | 10,890 |
| The Blade Itself | 0.66 | 0.29 | 12.55 | 1.14 | 0.24 | 103 | 107 | 6,769 |
| The Colour of Magic | 0.55 | 0.50 | 14.21 | 1.12 | 0.42 | 139 | 34 | 1,454 |
| The Gunslinger | **0.78**[°] | 0.25 | 13.43 | 1.11 | **0.17**[†] | 230 | 35 | 1,159 |
| The Lies of Locke Lamora | **0.21**[†] | 0.09 | **16.90**[°] | **1.38**[°] | 0.77 | 305 | 105 | 6,477 |

(Continued)

**Classic**

| Title | Fraction of defaults | Fraction of unidentified characters | Average sentence length | Average persons per sentence | Fraction of sentences with a person | Annotated sentences | Unique characters | Total character mentions |
|---|---|---|---|---|---|---|---|---|
| The Name of the Wind | 0.45 | 0.10 | 12.98 | 1.14 | 0.45 | 310 | 137 | 6,405 |
| The Painted Man | 0.30 | 0.28 | 14.67 | 1.29 | 0.70 | 301 | 137 | 9,048 |
| The Way of Kings | 0.31 | 0.29 | 12.20 | 1.10 | 0.36 | 316 | 221 | 14,696 |
| The Wheel of Time | 0.40 | 0.21 | 14.96 | 1.31 | 0.59 | **499°** | 188 | 9,426 |
| Way of Shadows | 0.32 | 0.13 | 13.53 | 1.32 | 0.56 | **88[†]** | 160 | 8,721 |
| Mean μ | 0.48 | 0.23 | 13.69 | 1.17 | 0.47 | 282.65 | 109.60 | 6,338.30 |
| Standard Deviation σ | 0.18 | 0.17 | 1.54 | 0.11 | 0.20 | 110.52 | 72.98 | 4,535.60 |
| $\mu_{classic} - \mu_{modern}$ | 0.08 | 0.05 | 7.61 | 0.04 | 0.01 | 7.45 | 16.00 | −1,383.65 |
| Pooled σ | 0.20 | 0.17 | 2.46 | 0.24 | 0.25 | 125 | 119 | 4,473 |
| $p$-Value | 0.21 | 0.39 | >0.01 | 0.73 | 0.74 | 0.85 | 0.68 | 0.35 |
| Significant | No | No | Yes | No | No | No | No | No |

**Note:**
The highest scores in each column are highlighted with a ◇, and the lowest scores with a †. The highest and lowest performing books for each class, in terms of $F_1$-score found in Tables A3 and A4, are marked with a grey fill. Boldface indicate the highest and lowest scores in each column.

**Table A4 Results of the complete BookNLP pipeline: Named entity recognition (Stanford NER), Character name clustering (e.g. 'Tom', 'Tom Sawyer', 'Mr. Sawyer', 'Thomas Sawyer' → TOM_SAWYER) and Pronominal coreference resolution.**

| Classic | | | | Modern | | | |
|---|---|---|---|---|---|---|---|
| Title | Precision | Recall | $F_1$-score | Title | Precision | Recall | $F_1$-score |
| 1984 | 77.33 | 72.87 | 75.03 | A Game of Thrones | 51.40 | 45.88 | 48.49 |
| A Study in Scarlet[○] | 40.00 | 37.22 | 38.56 | Assassin's Apprentice[○] | 37.00 | 34.89 | 35.91 |
| Alice in Wonderland | 54.93 | 48.36 | 51.43 | Elantris | 72.33 | 73.75 | 73.03 |
| Brave New World | 55.00 | 53.57 | 54.28 | Gardens of the Moon | 12.67 | 14.00 | 13.30 |
| David Copperfield[○] | 38.52 | 37.82 | 38.16 | Harry Potter | **79.17°** | **77.78°** | **78.47°** |
| Dracula[○] | 36.67 | 40.00 | 38.26 | Magician | 35.42 | 28.89 | 31.82 |
| Emma | **86.62°** | **86.50°** | **86.56°** | Mistborn | 61.99 | 60.62 | 61.30 |
| Frankenstein[○] | 51.16 | 45.35 | 48.08 | Prince of Thorns | 69.44 | 70.83 | 70.13 |
| Huckleberry Finn | 82.38 | 82.82 | 82.60 | Storm Front[○] | 40.54 | 39.19 | 39.85 |
| Jekyll and Hyde | 52.86 | 50.00 | 51.39 | The Black Company[○] | **6.85[†]** | **5.71[†]** | **6.23[†]** |
| Moby Dick[○] | 60.98 | 57.72 | 59.31 | The Black Prism | 76.90 | 77.59 | 77.24 |
| Oliver Twist | 77.64 | 74.35 | 75.96 | The Blade Itself | 34.09 | 36.36 | 35.19 |
| Pride and Prejudice | 73.55 | 72.22 | 72.88 | The Colour of Magic | 30.77 | 27.56 | 29.08 |
| The Call of the Wild | 30.00 | 25.19 | 27.38 | The Gunslinger | 77.84 | 75.89 | 76.85 |
| The Count of Monte Cristo | 40.72 | 35.80 | 38.10 | The Lies of Locke Lamora | 62.77 | 59.16 | 60.91 |
| The Fellowship of the Ring | 63.23 | 60.61 | 61.90 | The Name of the Wind | 61.38 | 58.67 | 60.00 |
| The Three Musketeers | **13.91[†]** | **12.17[†]** | **12.99[†]** | The Painted Man | 60.16 | 57.83 | 58.97 |
| The Way We Live Now | 66.07 | 66.79 | 66.43 | The Way of Kings | 65.87 | 64.42 | 65.14 |
| Ulysses | 66.67 | 66.98 | 66.82 | The Wheel of Time | 29.60 | 24.33 | 26.70 |
| Vanity Fair | 72.57 | 68.63 | 70.54 | Way of Shadows | 54.05 | 45.95 | 49.67 |

| Classic | | | | Modern | | | |
|---|---|---|---|---|---|---|---|
| **Title** | **Precision** | **Recall** | **$F_1$-score** | **Title** | **Precision** | **Recall** | **$F_1$-score** |
| Mean μ | 57.04 | 54.75 | 55.83 | Mean μ | 51.01 | 48.96 | 49.91 |
| Standard Deviation σ | 19.28 | 19.68 | 19.47 | Standard Deviation σ | 21.49 | 21.95 | 21.72 |

**Note:**
The highest scores in each column are highlighted with a ◇, and the lowest scores with a †. Novels written in 1st person are marked with a ⊙. Boldface indicate the highest and lowest scores in each column.

**Table A5 Social network measures for classic and modern novels.**

**Classic**

| Title | Nodes | Edges | Average degree | Average weighted degree | Network diameter | Graph density | Modularity | Connected components | Average clustering coefficient | Average path length |
|---|---|---|---|---|---|---|---|---|---|---|
| 1984 | 26 | 43 | 3.30 | 16.84 | 4 | 0.13 | 0.23 | 3 | 0.5 | 2.06 |
| A Study in Scarlet | 24 | 41 | 3.41 | 7.25 | 5 | 0.14 | 0.42 | 2 | 0.63 | 2.37 |
| Alice in Wonderland | 12 | 10† | 1.66† | 3.83† | 3 | 0.15 | 0.15 | 2 | 0† | 1.93 |
| Brave New World | 39 | 65 | 3.33 | 9.79 | 6 | 0.09 | 0.34 | 2 | 0.68 | 2.53 |
| David Copperfield | 142 | 499 | 7.03 | 23.11 | 6 | 0.05 | 0.49 | 2 | 0.57 | 2.69 |
| Dracula | 55 | 124 | 4.51 | 18.29 | 6 | 0.08 | 0.12† | 4 | 0.52 | 2.53 |
| Emma | 72 | 403 | 11.19 | 57.53◇ | 4 | 0.16 | 0.14 | 1 | 0.67 | 2.16 |
| Frankenstein | 20 | 38 | 3.80 | 10.60 | 5 | 0.20 | 0.51 | 2 | 0.75 | 2.41 |
| Huckleberry Finn | 62 | 121 | 3.90 | 8.42 | 7 | 0.06 | 0.52◇ | 4 | 0.60 | 3.30 |
| Jekyll and Hyde | 10† | 21 | 4.20 | 14.60 | 2† | 0.47◇ | 0.12 | 1 | 0.81◇ | 1.53† |
| Moby Dick | 90 | 169 | 3.76 | 7.38 | 8 | 0.04 | 0.44 | 8 | 0.59 | 3.33◇ |
| Oliver Twist | 62 | 191 | 6.16 | 22.32 | 4 | 0.10 | 0.32 | 2 | 0.75 | 2.26 |
| Pride and Prejudice | 62 | 373 | 12.03 | 57.10 | 4 | 0.20 | 0.16 | 1 | 0.73 | 1.96 |
| The Call of the Wild | 23 | 44 | 3.83 | 10.00 | 6 | 0.17 | 0.46 | 1 | 0.62 | 2.46 |
| The Count of Monte Cristo | 228 | 799 | 7.01 | 24.05 | 7 | 0.03 | 0.40 | 3 | 0.56 | 2.88 |
| The Fellowship of the Ring | 105 | 260 | 4.95 | 11.51 | 6 | 0.05 | 0.29 | 2 | 0.63 | 2.73 |
| The Three Musketeers | 96 | 279 | 5.81 | 15.33 | 5 | 0.06 | 0.32 | 1 | 0.55 | 2.56 |
| The Way We Live Now | 135 | 630 | 9.33 | 39.17 | 5 | 0.07 | 0.36 | 3 | 0.69 | 2.43 |
| Ulysses | 522◇ | 4,116◇ | 15.77◇ | 18.59 | 9◇ | 0.03 | 0.45 | 10◇ | 0.60 | 3.02 |
| Vanity Fair | 342 | 1,349 | 7.89 | 22.73 | 7 | 0.02† | 0.37 | 1 | 0.63 | 2.72 |
| Mean μ | 106 | 479 | 6.14 | 20 | 5.45 | 0.12 | 0.33 | 2.75 | 0.60 | 2.49 |
| Standard Deviation σ | 126.94 | 916.66 | 3.56 | 14.99 | 1.70 | 0.10 | 0.14 | 2.39 | 0.17 | 0.44 |

**Modern**

| Title | Nodes | Edges | Average degree | Average weighted degree | Network diameter | Graph density | Modularity | Connected components | Average clustering coefficient | Average path length |
|---|---|---|---|---|---|---|---|---|---|---|
| A Game of Thrones | 314◇ | 1,648◇ | 10.50◇ | 22.46 | 6 | 0.03 | 0.48 | 1 | 0.54 | 2.81 |
| Assassin's Apprentice | 55 | 110 | 4.00 | 9.09 | 6 | 0.07 | 0.34 | 2 | 0.49 | 2.65 |
| Elantris | 106 | 493 | 9.30 | 43.25◇ | 5 | 0.09 | 0.36 | 1 | 0.67 | 2.22† |
| Gardens of the Moon | 88 | 257 | 5.84 | 10.84 | 8 | 0.07 | 0.42 | 1 | 0.48 | 2.93 |
| Harry Potter | 67 | 198 | 5.9 | 19.37 | 5 | 0.09 | 0.15 | 1 | 0.68 | 2.23 |
| Magician | 84 | 209 | 4.98 | 10.76 | 6 | 0.06 | 0.43 | 2 | 0.58 | 2.83 |

(Continued)

**Classic**

| Title | Nodes | Edges | Average degree | Average weighted degree | Network diameter | Graph density | Modularity | Connected components | Average clustering coefficient | Average path length |
|---|---|---|---|---|---|---|---|---|---|---|
| Mistborn | 89 | 255 | 5.73 | 33.89 | 6 | 0.07 | **0.04**[†] | 3 | 0.62 | 2.37 |
| Prince of Thorns | 59 | 111 | 3.76 | 6.98 | 6 | 0.07 | 0.37 | 2 | **0.42**[†] | 2.83 |
| Storm Front | 33 | 85 | 5.15 | 10.97 | 4[†] | **0.16**[◊] | 0.31 | 1 | 0.64 | 2.26 |
| The Black Company | 30 | 45 | **3.00**[†] | **6.13**[†] | 6 | 0.10 | 0.20 | 3 | 0.561 | 2.43 |
| The Black Prism | 84 | 239 | 5.69 | 30.74 | 5 | 0.07 | 0.22 | 1 | **0.75**[◊] | 2.27 |
| The Blade Itself | 96 | 259 | 5.40 | 14.23 | 5 | 0.06 | 0.51 | 3 | 0.51 | 2.65 |
| The Colour of Magic | 27[†] | 43[†] | 3.19 | 7.93 | 6 | 0.12 | 0.38 | 1 | 0.50 | 2.67 |
| The Gunslinger | 31 | 69 | 4.45 | 8.52 | 7 | 0.15 | 0.41 | 1 | 0.43 | 2.87 |
| The Lies of Locke Lamora | 101 | 261 | 5.17 | 22.24 | 5 | 0.05 | 0.18 | 4 | 0.64 | 2.46 |
| The Name of the Wind | 109 | 197 | 3.62 | 8.99 | 9[◊] | 0.03 | **0.67**[◊] | 5 | 0.46 | **4.06**[◊] |
| The Painted Man | 132 | 444 | 6.73 | 23.15 | 7 | 0.05 | 0.53 | 1 | 0.63 | 2.70 |
| The Way of Kings | 172 | 448 | 5.21 | 20.79 | 6 | **0.03**[†] | 0.57 | 9[◊] | 0.55 | 2.91 |
| The Wheel of Time | 167 | 545 | 6.53 | 16.66 | 7 | 0.04 | 0.35 | 3 | 0.55 | 2.84 |
| Way of Shadows | 145 | 441 | 6.08 | 22.14 | 6 | 0.04 | 0.46 | 4 | 0.61 | 2.71 |
| Mean $\mu$ | 99 | 317 | 5.50 | 17 | 6.05 | 0.07 | 0.36 | 2.45 | 0.56 | 2.68 |
| Standard Deviation $\sigma$ | 66.37 | 348.92 | 1.85 | 10.05 | 1.15 | 0.04 | 0.15 | 1.99 | 0.09 | 0.4 |
| $\mu_{classic} - \mu_{modern}$ | 7 | 162 | 0.64 | 3 | −0.60 | 0.05 | −0.03 | 0.30 | 0.04 | −0.19 |
| Pooled $\sigma$ | 101 | 695 | 2.83 | 12.83 | 1.45 | 0.08 | 0.15 | 2.18 | 0.13 | 0.43 |
| $p$-Value | 0.83 | 0.47 | 0.49 | 0.55 | 0.20 | 0.09 | 0.42 | 0.67 | 0.37 | 0.17 |
| Significant | No | No | No | No | No | No | No | No | No | No |

Note:
The highest scores in each column are highlighted with a ◊, and the lowest scores with a †. The highest and lowest performinag books for each class, in terms of $F_1$-score found in Tables 3 and 4, are marked with a grey fill. Boldface indicate the highest and lowest scores in each column.

### Funding
The authors received no funding for this work.

### Competing Interests
The authors declare that they have no competing interests.

### Author Contributions
- Niels Dekker conceived and designed the experiments, performed the experiments, analysed the data, contributed reagents/materials/analysis tools, prepared figures and/or tables, performed the computation work, authored or reviewed drafts of the paper, approved the final draft.
- Tobias Kuhn contributed reagents/materials/analysis tools, authored or reviewed drafts of the paper, approved the final draft.

- Marieke van Erp conceived and designed the experiments, contributed reagents/ materials/analysis tools, authored or reviewed drafts of the paper, approved the final draft.

## Data Availability

Code and data are available at GitHub: https://github.com/Niels-Dekker/Out-with-the-Old-and-in-with-the-Novel.

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
