# Peer review of "Evaluating named entity recognition tools for extracting social networks from novels"

_PeerJ Computer Science, doi:10.7717/peerj-cs.189_

## Round 0.1 · original submission · Major Revisions

Thank you for submitting your manuscript to PeerJ Computer Science. All referees have completed the review of your manuscript and a summary is atached below. The reviewers recommend reconsideration of your paper following major revision. I invite you to resubmit your manuscript after addressing all reviewer comments.

Reviewer 1 ·

Basic reporting

- The article have many problems with English language. The English is very basic. Wrong expressions in many parts of the document.
- There are many incoherent or ambiguous expressions, and therefore they can not be understood throughout the article.
- Some ideas are very long, it would be necessary to use punctuation marks to separate them to better understand such ideas.
- The proposal must be explained more clearly in both the abstract and the introduction. After reading the following sections, the proposal is better understood.
- Some terms or methods need to be explained briefly. For example “small-world network”, “metaCPAN”, “gazetteers” among others.
- The authors should discuss related works that used networks to analyze texts. For example:
* doi.org/10.1209/0295-5075/99/28002
* doi.org/10.1371/journal.pone.0118394
* doi.org/10.1002/cplx.20305 
* doi.org/10.1371/journal.pone.0170527
- The figures and the results are good and relevant. There are quite a few results that have been shown and compared. However some results of the figures or tables need to be explained with a little more detail.
- The article is generally well structured, except in Section 5. For example, such a section contains two subsections with very similar names (network analysis and social network analysis), which could be put together into a single subsection. It also includes statistics that do not correspond to such subsection.
- Conclusion of basic reporting: Many modifications throughout the text are necessary to improve the readability and English used in the article

Experimental design

- The research of the article satisfies the Aims and Scope of the journal
- The article contains two research questions, but they are not well formulated. They need to be a little more specific and detailed.
- The knowledge gap is well established, indicating what will be the contributions of the article to cover this knowledge gap.
- The authors mentioned that there are multiple works of analysis of classical or ancient literature, however there are no works that make an analysis of modern literature. In the article they make several tests comparing both types of literature (classic vs modern), applying techniques of character recognition and social network analysis for both cases.
- The related works are well structured, indicating the most important aspects that should be taken into account for the development of the proposal research.
- The methods are quite clear and well detailed. Only that it would be necessary to give a better organization for the description of the method and obtained results.
- It might be convenient to separate a section that best explains the methodology, and in another section to include the explanation of all the obtained results.
- Conclusion of experimental design: OK, it is necessary to better organize some sections, and to make the research question clearer.

Validity of the findings

- The article showed a large number of results, comparing the two types of literature proposed (classical literature vs. modern literature)
- Different character identifiers were compared for the two types of literature. Additionally several tests were done with these methods.
- There were comparisons regarding the measures or features from networks generated for both types of literature.
- There were comparisons on overall measures for both types of literature
- An analysis was made about possible improvements to common problems that can be found in the task of character recognition.
- The selection of books that were chosen for each type of literature (classic vs modern literature) was explained in detail.
- Comparison of 20 classic books and 20 modern books. However, only the first or the first chapters of each book were taken into account. Is it relevant? Base this choice
- Is the comparison of the network measures for these books relevant? Not much detail was explained about the network measures. Only one explanation was given in a very general way.
- The behavior of two networks was explained visually (one book for each type of literature): some general details and possible drawbacks of these networks were given. It would be interesting to give a more general explanation of all or most of the networks generated for each book. Would a visual analysis of each network be possible? A brief explanation would be relevant.
- The conclusion is not very clear. The conclusions need to be better explained based on the results obtained. It is also not clear due to several problems with English language.
- The authors encourage to other researchers to do future works so that such works can reinforce some of the results derived from this research.
- Conclusion of validity of the findings: OK.

Additional comments

- The research article is good, it presents different relevant results, and it makes many comparisons. And also it gives suggestions for future work.
- However, it is necessary to review everything again, because there are many problems with English language. It is necessary to correct, because there are also enough sentences that can not be understood.
- In both the abstract and the introduction, it would be necessary to better explain the methodology of the work. Likewise, the research questions should be improved or be more specific.
- Regarding the corpus of selected books, it should be justified a little better why only the first chapters of the book were selected. Would the results change a lot if the whole books are analyzed?
- Give a better organization regarding the methodology and results obtained from network analysis
- All revisions are in the PDF document. Being red highlighted the errors concerning to English language erros; while green highlighted are the mistakes about things that need to be better explained. The main ideas found in the article are yellow highlighted.

Annotated reviews are not available for download in order to protect the identity of reviewers who chose to remain anonymous.

Reviewer 2 ·

Basic reporting

The manuscript entitled "Evaluating social network extraction for classic and modern
fiction literature" presents a study that provides a comparison between classic and modern literature through the use of natural language processing tools for the automatic extraction of social networks.

The manuscript is very well written and there are just a few typos and corrections to be performed in the text.
- line 45: missing space between "time" and "Biber"
- line 66: the second "analysis" should be removed
- line 98: the word "obtaining" is repeated
- line 99: the F-1 score should be 0.936?
- line 184: missing space between "removed" and "(1)"
- line 214: sentence needs correction "by the running the"

The literature references, the main background and the introduction of the theme are well presented, although the main motivation for extracting social networks from books is not well stated. This should be better explored to bring the reader's attention for what concerns the possible applications of such social networks.

The manuscript is well structured, presents a logical sequence of sections and a good summary of the available data and the obtained results. The data used in the manuscript is also available online through a link provided by the authors, except the books with copyright restrictions. However, the gold standard annotations of these books are also provided in the project's web page.

Experimental design

The purpose of the study described in the manuscript is well defined and the main topic is relevant to the scope of the journal. The methodology is also well described allowing the reproducibility of the work. As the main application of the study, the authors try quantify the differences and similarities between classic and recent fiction literature, which is a gap that has not been adressed yet by previous related works.

Validity of the findings

The authors provided an in-depth comparison of four Named Entity Recognition systems. They identified some issues that can be related to the performance of these systems in recognizing characters, such as real word names. They showed how performance can be increased by applying some transformations to the corpus which are able to handle those issues.

This study draws attention on how Networks can contribute as an analysis tool in the context of natural language processing. However, for what concerns the extraction of social networks from the novels, the quantitative analysis showed no significant differences between classic and modern literary books. I would like to highlight the following issues:

- The network measures are very different within each group. Given such differences in the number of nodes and the number of edges, it becomes difficult to compare these networks among themselves just by analysing the mean and standard deviation. Are the networks with such different degrees comparable to each other?

- Throughout the description of the results and their respective analysis, the authors raised some alternatives from the methodological point of view. For instance, the use of conversational networks instead co-occurrence networks and the use of a list of aliases. To which extent these alternatives can influence the conclusions about the differences and similarities between classic and modern books? This should be elaborated in the discussion or conclusion parts.

- No significant differences were found between the performance of classic and modern novels, and also between the two types of social networks. Would this result be beneficial from the point of view of network analysis, given for instance, the community structures and the social interactions described in the books?

- Did the authors consider the possibility of using machine learning techniques to distinguish both classes of literary books?

- In Table 10, the authors present the network measurements for only 18 books of the dataset for each class. Why two books were removed for each class in this table? In addition, the mean values presented for each measure do not correspond to the actual mean of each column. This table should be reviewed.

Reviewer 3 ·

Basic reporting

no comment.

Experimental design

Please see the detailed comments below. There is a research question, but the gap is not clearly defined. There is clearly a very rigorous investigation. The approach is clearly described, as could be replicated.

Validity of the findings

Please see the detailed comments below. There are some questions about the two selected cases, but I trust the method is sound. The discussion is lacking in specification, but I believe this can be improved in two ways: either reordering the argumentation, or developing two separate papers. I have detailed this in the comments below.

Additional comments

This paper reviews the NLP tools used to automatically identify literary networks. It reviews four popular and thus commonly implemented NER tools. It also generates networks from the results of NER from the tool BookNLP. This paper has a lot of potential, the authors have done a valid and systematic comparison of the tools, but there is a desperate need to clarify concepts, contributions and the argument, as to enhance this contribution. In addition, I would strongly advise the authors to consider developing this into two different papers to support the two differing yet complementary arguments that they attempt to make in the paper. One comparing the NLP tools for conducting NER and another of the pipeline from NLP tools of NER to networks and thus the network analysis of the literary networks. I think this would do the two arguments and contributions much more justice. For this review, I have assumed they will keep with the dual argument and have made a set of suggestions to improve the paper, with mentions when necessary as evidence for separate papers.
As I mentioned above, there is a desperate need to clarify concepts, contributions and the argument(s), as to enhance the contribution(s). For example: Already in the introduction the argument is lacking clarity which leaves the reader confused: first the authors say NLP is important and we need compare these tools (but fail to tell the reader why), then the authors state that actually the problem is that we do not have comparisons of the 18th + 19th century texts with modern texts as the styles are different and thus we cannot compare the networks (but still the reader does not know what literary network analysis tell us and why that would matter). And then the authors say we need to understand “the potential differences or similarities in terms of (1) performance of natural language processing techniques, (2) social network structure, and (3)overall writing style”. Then in your research question you state you are looking only at fantasy novels, but with no context as to how that is an element of the overarching research question. The reader cannot grasp what the main contribution is as the authors fail to present a logical argument and detail knowledge (what is currently now presented as assumptions). Please see this as an opportunity to clarify your points.
There is a great need to clarify your contribution and build an argument so that these three above points build upon each other. Currently they are just muddled through each other. Being much more precise about your boundaries is going to enhance your argument immensely. There are a number of arguments that need to be established, that have either been overlooked or assumed, where it would serve the authors to be much more precise:
First, in the first paragraph of the introduction the authors need to establish: what are literary networks and what can they tell us? (do not mention NLP yet). Please explain and cite what it is meant by literary networks. You seem to infer it is social networks and you reference Moretti in the first paragraph, but then in paragraph 3 it is stated that the concept and state of interest is the language and style that changes over time. Thus there is a step missing for the reader, it would serve one well to clarify your scope to identify how others have considered and identified links – are we talking about references between characters, across novels/works, etc? Also what does the mapping of literary networks serve to explain? I do know this elaborated upon in the section Related Work, but this is separate from these techniques, this has to be established as to understand why NER is being used as a technique for this approach.
In addition, as part of the second argument, the authors state later “we have not found any other studies that compared the performances of social network extraction on classic and modern novels; or compared the structures of these networks”, which gives the reader the assumption that this is important, but it has not been established in the introduction why this is important and why it is a gap that needs to be filled. Also note this gap is different than the one the authors attempt to establish later – comparing time periods. It is fine to have two contributions, but then please highlight them as two gaps and very precisely how they build upon each other. Again this question arises- could this potentially be two separate papers, which would allow you elaborate in much greater detail on both contributions.
Second, review the current tools that have been used to identify literary networks. Currently, indirectly the authors state that NLP is increasingly being used, and that there are multiple tools, but they do not directly establish why it is important to review them. You may want to state here that you are only focusing on computational tools, as to exclude a review of close reading methods. This review should summarize with your key argument (as presented now) that there is a need to compare networks over time. Please state why this is so important. I believe you attempt to do this, but not directly, you need to rehash the state of the art here.
Third mention the example, related to the periods. Do note: If you are going to use literary novels as your source and thus target literary scholars (as I am assuming you are), you need to get the terminology correct. 18th and 19th century texts are not classics. In popular understanding they may be largely referred to as classic/timeless novels that make up a canon of must read texts, but these are not classic texts. In literary studies, the classics refer to works from ancient Greece and Rome. This period you refer to, pending how far you go into the 19th century, can be referred is often as the Enlightenment – starting around 1660, with the crowning of the exiled Charles II, until the beginning of the 19th century and the reign of Victoria. Although it may also include the period of Romanticism (early 1800s to Victorian period). If you are largely speaking of English texts then you may consider to mention it as Victorian- under the reign of Queen Victoria (1837–1901). In any case you need to pick another term as it is incorrect to call this this classics when you are referring to literary studies. The use of modern is applicable, as if I understand it is looking at the 20th century. Although, you are going to have to justify later why you looked at fantasy novels as your modern set, and not canonical works as you did for the 18th and 19th century. Please see the comments related to fantasy as a selection.
Fourth, the research questions would be fine if it did not include the fantasy mention. The problem lies in that fantasy is a literary genre, not a period (as you mentioned before as your core gap) and classics as you refer to them are not a genre. These top novels of the 18th and 19th century are a canonical distinction. And one could argue fantasy is not related to your research question, but rather a methodological choice to limit even further the assumed variation between periods. Thus the fantasy bit can be included in the write up of the approach. On the other hand in selecting these two cases the authors have actually selected two incomparable cases: an overarching representation of 18th and 19th century novels and a specific genre from the 20th century. So either select for representativeness of a period or on genre or both. This is very important given that the authors state that they seek to contribute to the understanding of how to study or compare differences between these networks, but then select comparable cases from that has similar characteristics for the two cases. I suggest/guess, there must be another Guardian list of 20th or 21st century novels that makes up a canon.
Fifth, mention here the approach, how the research questions be answered. As is mentioned “The contributions of this paper are (1) an annotated gold standard dataset with entities and coreferences of 20 classic and 20 modern novels, (2) a comparison and an analysis of four named entity recognition on 20 classic and 20 modern novels, (3) a comparison and an analysis of social network analysis measures, and (4) experiments and recommendations for boosting performance on recognising entities in novels”. Explain shortly the steps of each of these.
Then begin the related work section. I can follow the steps made and the analysis. Although I think the authors can expand greatly on the discussion. This is particularly true in the case of the network measures, the authors measure a number of measures, show lots of visualisations, but to do them any justice. The author need to elaborate on why these measures, how do they compare, how do they contribute back to this gap. It is one thing to say, look we can do this, but given your initial framing for both technical NLP and literary researchers I would challenge you do more and expand on this.
In regards to analysis of the networks, it is very superficial. Since you aim to contribute to literary network analysis, it would be good to employ a network analyst who can assist on integrating the measures, methods, etc and to link to the contributions the authors attempt to make. For example, the authors have measured so many aspects of the network, let that terminology guide you. Some examples: very full network is not a network term; if you want to describe a full network measure density and describe the networks as a continuum from sparse and dense. There are assumptions such as: “the relationship between the main characters of this novel can easily be identified” – please explain how they are identified? And note which characters these are. Also a visualisation is not an analysis, it is a visualisation, one may be able identify clusters by the eye as suggested - “The visualisation of such a network also offers a prompt manner to identify social clusters.”, but to serve as evidence one needs to use graph theory and let the mathematically evidence serve as the evidence of clusters. Thus, if you want to discuss clusters use a clustering algorithm. Again, unfortunately, the authors are just selling themselves short by not maximizing the use of the network measures and terminology. Consequently, the discussion and contributions fall short, which is a pity.
Overall, I truly think the paper is very interesting and needed in this field; but the authors are not doing their work justice, by stuffing them into one paper. I think it would serve them better to develop two papers. One that makes a case for comparing tools and thus time periods and then a second for the network structures. This comes to light a number of times, both in the current lack of specificity of the argument and that in Section 5, the authors only generate the networks from one tool – BookNLP, which is a different initial approach to comparing the tools. In addition, I would argue that these contributions, although related, actually speak to different sub-sets of the research community- 1) using NER tools, and then 2) those that use NER tools for literary networks specifically and how to understand these structures. In developing two papers the authors can expand on their discussion of the findings for both NER tools and networks.
There are number of instances where you are missing commas, which I did not note here, as I am assuming this is related to the Latex file, please check accordingly.

---

## Round 0.2 · Minor Revisions

Please take into consideration all the referees' suggestions in order to further improve the quality and impact of your manuscript.

Reviewer 1 ·

Basic reporting

Several aspects suggested in the first revision were corrected, however there are still some errors or unclear pieces of text regarding the English writing of the manuscript. Some sentences need to be corrected because they still were not clear. In other cases, some sentences are very long, it would be necessary to use punctuation marks to separate them to better understand such sentences. The sentences could be found at lines:
- 57 to 59
- 106 to 107
- 113 to 114
- 129 to 130
- 133 to 134
- 155 to 156
- 189 to 190
- 240
- 283
- 305 to 306
- 310 to 312
- 318 to 319
- 414 to 416

Experimental design

no comment

Validity of the findings

no comment

Additional comments

The research article is good, it presents different relevant results, and it makes many comparisons. And also it gives suggestions for future work. The article showed a large number of results, comparing the two types of literature proposed (classical literature vs. modern literature).

The authors addressed all the points raised by the previous revision. However, it is required to make some changes related to the English in the text in some sentences. The sentences that need to be reviewed are indicated in the basic reporting section.

Reviewer 2 ·

Basic reporting

no comments

Experimental design

no comments

Validity of the findings

no comments

Additional comments

My main questions have been addressed by the authors. The performed changes resulted in a more clear and organized text.

Reviewer 3 ·

Basic reporting

This revision of the text is vastly improved, from the previous version. The methods, findings and discussion sections in particular, have improved in clarity, specifically that the approach is now explained in detail, and the decisions explicated to the research question. That being said, I still think that the positioning of the paper can be even more specific.

This increased specify begins with the first paragraph to the research questions, in the introduction. The very first paragraph attempts to position the study in a larger field, but in doing so it not only incorrectly positions the contribution and scope to making contributions to literary theory (which the paper does not do), but also casts the scope so large to quantification of literary studies; when in fact your study is much more specific contributing to knowledge specifically to character network identify through NLP techniques. As the Game of Thrones example proves and which you state on page 13 line 461 “ This is a design choice that [462] will depend on the type of research question one wants to answer by analyzing the social networks”. Yes! Yes! This is your contribution! – in order to understand the validity and reliability of these character networks and thus our understanding of social structures in novels, this starts from the nature that the nodes and edges were automatically identified using the NLP/NER tools. This is so fundamentally important, and your approach gives scholars great insights into which tool is best suited for their study or research question. Thus drop the entire framing of contributions to literary theory, claim your findings and their boundaries and roll with them!

Thus, I would suggest a re-positioning. This would mean that I would highly recommend removing the first paragraph completely, and instead start out with why the understanding of social networks in novels are important in literary studies. Then narrowing your scope to how these networks can be identified, and the NLP approaches differ and how. Thus you then have one more general research question: To what extend do NLP techniques using NER differ in identifying social networks in novels? / as to how to operationalization networks (really what is a node and edge) and how that translates to our understanding of the projection of the networks and thus analysis of the networks. Where you first look at the NER process and compare it, and then the network itself and its relationship to the extraction/identification.

In addition, this suggests that your framing of the importance and selection of the two time periods instead becomes part of your case selection, not part of your aim. It is not your aim to contribute to our understanding of different periods, you even say that no differences have been found and no other research has been done this (starting at page 2). Thus I would highly recommend you delete this claim/positioning. Instead mention the time periods and genres in the methods section as a variable you consider to insure that it is applicable to “all” periods; much like you compared these networks to real world networks. This serves to further ensure the validity of the findings, but it is actually irrelevant to the approach and contribution you aim to make.
This then suggests that the title becomes more general and thus more powerful! – evaluating social network extraction tools for fictional literature.

In addition, given one of your core audiences is literary scholars, I still adamantly oppose to you using the word classics, these are canonical works as you also mention now in a footnote. But no literary scholar is going to take your work seriously if you from the outset incorrectly categorize this core term in their field.

Experimental design

See the above comments.

Validity of the findings

See the above comments.

Additional comments

I want to emphasize, the paper has improved greatly, and these suggestions suggest a further specification of the contribution and message, that are minor revisions that should even further enhance the impact of this work.

---

## Round 0.3 · accepted · Accept

Please check and correct all remaining language issues.

Reviewer 1 ·

Basic reporting

Several aspects suggested in the previous revision were corrected, however, there are still some errors or unclear pieces of text regarding the English writing of the manuscript. The unclear sentences or some mistakes could be found at lines:
- 21-23: Furthermore, we identify several issues that complicate named entity recognition in our set of novels and we present methods to remedy these.
- 33-34 The most commonly used approach for extracting such networks, is to first identify characters …
- 136-137: One difficulty of character detection is the variety of aliases one character might go by, or; coreference resolution.
- 367: However, other than that, none of the other measures differ significantly …
- You must use a unique pattern to talk about F1-score measurements:
* 121: F1-scores between: .45 and .54. Vala et al. …
* 130: obtaining an F1 score of 0.936 for ...
* 308: yields a F1 score of 51.91. …
* 545-546: F1 score rose from 13.91 to 53 ...

Experimental design

no comment

Validity of the findings

no comment

Additional comments

The authors addressed most of the points raised by the previous revision. However, it is required to make some changes related to the English in the text in some sentences. The sentences that need to be reviewed are indicated in the basic reporting section.

Reviewer 2 ·

Basic reporting

no comments

Experimental design

no comments

Validity of the findings

no comments

Additional comments

The reviewer's suggestions concerning the contributions of the paper were incorporated in this second revision by the authors. The paper was greatly improved.

Reviewer 3 ·

Basic reporting

The paper has improved greatly in precision and clarity.

Experimental design

Sufficient.

Validity of the findings

Sufficient.

Additional comments

I recommend this paper for publication, and have no further questions or concerns.

---

## Author Rebuttal · Round 0.3

# Response to the Reviewers' Comments

Marieke van Erp, Niels Dekker, and Tobias Kuhn

Dear Editor and Reviewers,

We are happy that our previous revisions were largely satisfactory to the reviewers and we are grateful for the additional comments and suggestions to improve our article. We have revised our manuscript accordingly. We have addressed the typos and missing punctuation marks spotted by Reviewer 1 and proofread the text again. Furthermore, we have followed Reviewer 3's comments on strengthening the positioning of our work. Please find detailed answers regarding these modifications below.

## I. DETAILED RESPONSE TO REVIEWER 3

> [...] I still think that the positioning of the paper can be even more specific.
> This increased specify begins with the first paragraph to the research questions,
> in the introduction. The very first paragraph attempts to position the study in a
> larger field, but in doing so it not only incorrectly positions the contribution
> and scope to making contributions to literary theory (which the paper does not
> do), but also casts the scope so large to quantification of literary studies; when
> in fact your study is much more specific contributing to knowledge specifically to
> character network identify through NLP techniques. As the Game of Thrones example
> proves and which you state on page 13 line 461 ``This is a design choice that
> [462] will depend on the type of research question one wants to answer by
> analyzing the social networks". Yes! Yes! This is your contribution!  in order to
> understand the validity and reliability of these character networks and thus our
> understanding of social structures in novels, this starts from the nature that the
> nodes and edges were automatically identified using the NLP/NER tools. This is so
> fundamentally important, and your approach gives scholars great insights into
> which tool is best suited for their study or research question. Thus drop the
> entire framing of contributions to literary theory, claim your findings and their
> boundaries and roll with them!

**Response 1.** Thank you for your kind words and insightful analysis of our work. We have amended the introduction to draw more attention to the motivation behind evaluating the robustness of automatic extraction methods in order to know how they behave on different kinds of texts.

> Thus, I would suggest a re-positioning. This would mean that I would highly
> recommend removing the first paragraph completely, and instead start out with why
> the understanding of social networks in novels are important in literary studies.
> Then narrowing your scope to how these networks can be identified, and the NLP
> approaches differ and how. Thus you then have one more general research question:
> To what extend do NLP techniques using NER differ in identifying social networks
> in novels? / as to how to operationalization networks (really what is a node and
> edge) and how that translates to our understanding of the projection of the
> networks and thus analysis of the networks. Where you first look at the NER
> process and compare it, and then the network itself and its relationship to the
> extraction/identification.

**Response 2.** We have removed the first paragraph and rewritten the second in order to bring forward the focus on the analysis of NLP techniques.

> In addition, this suggests that your framing of the importance and selection of
> the two time periods instead becomes part of your case selection, not part of your
> aim. It is not your aim to contribute to our understanding of different periods,
> you even say that no differences have been found and no other research has been
> done this (starting at page 2). Thus I would highly recommend you delete this
> claim/positioning. Instead mention the time periods and genres in the methods
> section as a variable you consider to insure that it is applicable to ''all"
> periods; much like you compared these networks to real world networks. This serves
> to further ensure the validity of the findings, but it is actually irrelevant to
> the approach and contribution you aim to make.

**Response 3.** We selected the two periods because we realised that in prior work we had not and have not yet seen these types of analyses done on contemporary literature and one of our questions initially was whether the tools would behave differently. The fact that we did not find these differences in our opinion does not preclude us from explaining this in the introductory section.

We have amended our text to put less stress on the comparison between the two periods, but still mention it and keep the distinction between the two corpora, as we deem it important that the corpora on which NLP tools are tested are as diverse as possible.

> This then suggests that the title becomes more general and thus more powerful!
> evaluating social network extraction tools for fictional literature.

**Response 4.** Thank you for your suggestion, we have updated the title to reflect the focus on the evaluation of the extraction more.

> In addition, given one of your core audiences is literary scholars, I still
> adamantly oppose to you using the word classics, these are canonical works as you
> also mention now in a footnote. But no literary scholar is going to take your work
> seriously if you from the outset incorrectly categorize this core term in their
> field.

**Response 5.** As we state in the manuscript, as well as in our previous response letter, we are following Charles Augustin Sainte-Beuve in our nomenclature. Furthermore, we do not refer to our corpus of 19th century novels 'classics' but 'classic novels'. We also consulted with a literature studies professor at one of our institutes who did not object to this. We therefore think that we sufficiently distinguish our dataset from the ancient Greek and Roman classics.